



# Lability classification of soil organic matter in the northern permafrost region

*Kuhry, Peter[1], Bárta, Jiří[2], Blok, Daan[3], Elberling, Bo[4], Faucherre, Samuel[4], Hugelius, Gustaf[1,5], Richter, Andreas[6], Šantrůčková, Hana[2] and Weiss, Niels[1,7]*

[1] *Department of Physical Geography, Stockholm University, Sweden*
[2] *Department of Ecosystem Biology, University of South Bohemia, Ceske Budejovice, Czech Republic*
[3] *Department of Physical Geography and Ecosystem Science, Lund University, Sweden*
[4] *CENPERM, University of Copenhagen, Denmark*
[5] *Bolin Centre for Climate Research, Stockholm University, Sweden*
[6] *Department of Microbiology and Ecosystem Science, University of Vienna, Austria*
[7] *Current affiliation: Department of Geography and Environmental Studies, Carleton University, Ottawa, Canada*

*Correspondence email: peter.kuhry@natgeo.su.se*

Abstract

The large stocks of soil organic carbon (SOC) in soils and deposits of the northern permafrost region are sensitive to global warming and permafrost thawing. The potential release of this carbon (C) as greenhouse gases to the atmosphere does not only depend on the total quantity of soil organic matter (SOM) affected by warming and thawing, but also on its lability (i.e. the rate at which it will decay). In this study we develop a simple and robust classification scheme of SOM lability for the main types of soils and deposits of the northern permafrost region. The classification is based on widely available soil geochemical parameters and landscape unit classes, which makes it useful for upscaling to the entire northern permafrost region. We have analyzed the relationship between C content and C-$CO_2$ production rates of soil samples in two different types of laboratory incubation experiment. In one experiment, c. 240 soil samples from four study areas were incubated using the same protocol (at 5 °C, aerobically) over a period of one year. Here we present C release rates measured on day 343 of incubation. These long-term results are compared to those obtained from short-term incubations of c. 1000 samples (at 12 °C, aerobically) from an additional three study areas. In these experiments, C-$CO_2$ production rates were measured over the first four days of incubation. We have focused our analyses on the relationship between C-$CO_2$ production per gram dry weight per day ($\mu gC$-$CO_2$ gdw$^{-1}$ d$^{-1}$) and C content (%C of dry weight) in the samples, but show that relationships are consistent when using C/N ratios or different production units such as $\mu gC$ per gram soil C per day ($\mu gC$-$CO_2$ gC$^{-1}$ d$^{-1}$) or per cm$^3$ of soil per day ($\mu gC$-$CO_2$ cm$^{3-1}$ d$^{-1}$). C content of the samples is positively correlated to C-$CO_2$ production rates but explains less than 50 % of the observed variability when the full datasets are considered. A partitioning of the data into landscape units greatly reduces variance and provides consistent results between incubation experiments. These results indicate that relative SOM lability decreases in the order: Late Holocene eolian deposits > alluvial deposits and mineral upland soils (including peaty wetlands) > Pleistocene Yedoma deposits > C-enriched pockets in cryoturbated soils > peat deposits. Thus, three of the most important SOC storage classes in the northern permafrost region (Yedoma, cryoturbated soils and peatlands) show low relative SOM lability. Previous research has suggested that SOM in these pools is relatively undecomposed and the reasons for the observed resistance to decomposition in our experiments needs urgent attention if we want to better constrain the magnitude of the thawing permafrost carbon feedback on global warming.





1. Introduction

Permafrost has been recognized as one of the vulnerable carbon (C) pools in the Earth System (Gruber et al., 2004). In the most recent decade there has been a surge in papers dealing with the permafrost carbon feedback on climate change (e.g. Schuur et al., 2008; Kuhry et al., 2010). This increased interest was fueled by a new and high estimate of the total soil organic carbon (SOC) storage in the northern permafrost region (Tarnocai et al., 2009), which was received with great interest by the Earth System Science community (e.g., Ciais, 2009). Since this first new estimate was published, a multitude of new SOC inventories at the landscape level have been conducted across the Circumpolar North (e.g. Hugelius and Kuhry, 2009; Hugelius et al., 2010; Horwath Burnham and Sletten, 2010; Palmtag et al., 2015; Gentsch et al., 2015; Siewert et al., 2016). Recent studies have also focused on re-evaluating the spatial extent and SOC storage of the Yedoma 'Ice Complex' and Alas deposits (Strauss et al., 2013; Walter-Anthony et al., 2014; Hugelius et al., 2016; Shmelev et al., 2017).

This new data has prompted an update of the total SOC storage in the northern permafrost region, its vertical partitioning and its broad (continental scale) distribution (Hugelius et al., 2014). The new estimate amounts to c. 1400 PgC for the top 3 m of soils and deeper deposits, including permafrost and non-permafrost organic soils (Histels/Histosols, 302 PgC), cryoturbated permafrost mineral soils (Turbels, 476 PgC), non-cryoturbated permafrost mineral soils (Orthels) and non-permafrost mineral soils (256 PgC), and deeper Yedoma (301 PgC, >300 cm) and Delta (91 PgC, >300 cm) deposits. The spatial distribution of SOC stocks according to the major permafrost soil (Gelisol) suborders, non-permafrost mineral soils and Histosols (Soil Survey Staff, 2010) is graphically represented in the updated version of the Northern Circumpolar Soil Carbon Database (NCSCDv2, 2014).

The importance of an accurate estimate of total SOC storage in the northern permafrost region is illustrated by a recent review of the permafrost carbon feedback (Schuur et al., 2015), which included a comparison of future C release in a total of eight Earth System models (ESMs). The magnitude of the projected cumulative C loss from thawing permafrost by 2100, largely based on the RCP 8.5 scenario (IPCC, 2013), varied greatly between models from 37 to 174 PgC. However, by normalizing for the initial permafrost C pool size in the different ESMs, the proportional C loss from the permafrost zone was constrained to a much narrower range of $15 \pm 3$ % of the initial pool. This indicates that the quantity of SOC is a primary control when assessing C losses from the northern permafrost region.

The magnitude of the permafrost carbon feedback, however, will not only depend on the rate of future global warming (and its polar amplification), its effect on gradual and abrupt permafrost thawing (Grosse et al., 2011), or the total size (and vertical distribution) of the permafrost SOC pool. As shown by Burke et al. (2012), based on simulations with the Hadley Centre climate model, quality (decomposability) parameters need also to be considered. Thus, in terms of C pool parameters, the potential C release from the northern permafrost region will depend not only on SOC quantity but also on soil organic matter lability (i.e. the rate at which SOM will decay following warming and thawing). An important tool to assess potential C release from permafrost soils and deposits are laboratory incubation experiments that consider both different types of substrate (e.g., Schädel et al., 2013) and time of incubation (e.g. Elberling et al., 2013).

The aim of this study is to add a measure of SOM lability to the current estimates of SOC quantity, in order to define vulnerable C pools across the northern circumpolar region. We focus on the relationship between solid phase geochemical parameters (particularly C content) and C release rates in laboratory incubations of active layer and thawed permafrost samples from the main types of soils and deposits found in the northern permafrost region. Our objective is to develop a SOM lability classification scheme based on widely reported soil geochemical parameters in field SOC



inventories and general landscape classes, that can be linked to existing spatial SOC databases such
as the NCSCD (Tarnocai et al., 2009; Harden et al., 2012; Hugelius et al., 2014). We test the
robustness of our SOM lability classification by comparing two very different types of incubation
experiment, both in setup as well as timing of C release measurements.

2. Materials and methods

*2.1. Study areas*

The samples used in the incubation experiments were collected as part of landscape-level inventories
carried out in the context of the EU PAGE21 and ESF CryoCarb projects to assess total storage,
landscape partitioning and vertical distribution of SOC stocks in study areas across the northern
permafrost region. SOC storage data from these areas are published in Weiss et al. (2017) for
Svalbard, Siewert et al. (2016) for Lena Delta, Palmtag et al. (2016) for Taymyr Peninsula, Palmtag
et al. (2015) for Lower Kolyma, Hugelius et al. (2011) for Seida, and Siewert (2018) for Stordalen
Mire. The location of all study areas is shown in Fig. 1. The Lower Kolyma experiment includes
samples from two nearby located study areas (Shalaurovo and Cherskij); the Taymyr Peninsula
experiment also includes samples from two nearby located study areas (Ary-Mas and Logata).
Metadata for each of these areas, including geographic coordinates, permafrost and vegetation zones,
climate parameters, number of soil profiles and incubated samples, type of incubation experiment,
and time of field collection, are presented in Table S1 (Supplementary Materials).

Figure 1. Location of study areas in northern Eurasia. PAGE21 experiment (Ny Ålesund,
Adventdalen, Stordalen Mire, Lena Delta); CryoCarb 1-Kolyma experiment (Shalaurovo,
Cherskij); CryoCarb 2-Taymyr experiment (Ary-Mas, Logata); CryoCarb 3-Seida experiment.
Permafrost zones according to Brown et al. (1997).


*2.2. Field methods*

The sampling strategy applied for SOC field inventories was aimed at capturing all major landscape
units in each of the study areas, while at the same time ensuring an unbiased selection of soil profile
location. This semi-random sampling approach consisted of deciding on the positioning of generally
1 or 2 km long transects that crossed all major landscape units, with a strictly equidistant sampling
interval at normally 100 or 200 m that eliminated any subjective criteria for the exact location of
each soil profile. For SOC storage calculations, the mean storage in each landscape unit class was
weighed by its proportional representation in the study area based on remote sensing land cover
classifications.
At each soil profile site, the topsoil organic layer was collected by cutting out blocks of known
volume in three random replicates to account for spatial variability. These samples do not always
strictly adhere to the definition of an 'O' soil genetic horizon, because in areas with thin topsoil
organics (like in floodplains and mountain terrain) there can be a large admixture of minerogenic
material resulting in C contents of less than 12 %. Active layer samples were collected from
excavated pits by horizontally inserting fixed-volume cylinders. The permafrost layer was sampled
by hammering a steel pipe of known diameter incrementally into the ground, retrieving intact
samples for each depth interval. Depths intervals are normally 5 to 10 cm or less (e.g., when the
topsoil organic layer was very thin). The standard sampling depth was 1 m below the soil surface; at





some sites it was not possible to reach this depth due to large stones in the soil matrix or thin soil
overlying bedrock (often in mountainous settings).

*2.3. Incubation experiments*

**2.3.1. The PAGE21 incubation experiment**

The PAGE21 incubation experiment was carried out at the University of Copenhagen (Denmark).
This experiment included one sample from the topsoil organics, one sample from the middle of the
active layer and one sample from the upper permafrost layer (normally 10-15 cm below the upper
permafrost table) from all mineral soil profiles collected in three of the PAGE21 study areas.
Samples were selected based on depth criteria and not any specific soil characteristic (e.g., presence
of C-enriched cryoturbated material or absence of excess ground ice). In some cases, upper
permafrost samples could not be collected due to very deep active layers and/or thin soils
(particularly in mountain settings). Peat samples are available from a fourth PAGE21 study area. In
total c. 240 soil samples from four study areas across the northern permafrost region (Ny Ålesund
and Adventdalen, Svalbard; Stordalen Mire, N Sweden; Lena Delta, N Siberia) were incubated in
one and the same experiment (Faucherre et al., 2018).
The Dry Bulk Density (DBD) of samples used for incubation was measured at Stockholm
University (Sweden). The %C and %N of dry weight of the incubated samples were measured in an
elemental analyzer (EA Flash 2000, Thermo Scientific, Bremen, Germany) at the University of
Copenhagen (Denmark).
Samples were kept in frozen condition from collection until the start of the laboratory incubation
experiment. Samples were incubated at 5 °C and field water content levels (aerobic conditions) over
a one-year time period. Mean volumetric water content varied between 30 % (topsoil organics), 45-
50 % (active layer and permafrost layer mineral soil) and 69 % (peat). C-$CO_2$ production rates were
measured at five different occasions between 7 to 343 days after the start of the experiment, using a
nondispersive infrared LI-840A $CO_2$/$H_2O$ Gas analyzer (LICOR® Biosciences). Here, we use C
release rates after nearly one year of incubation as a measure of SOM lability. Since all samples from
all study areas were processed and incubated using the same protocol, results are directly
comparable.

**2.3.2. The CryoCarb incubation experiments**

The CryoCarb incubations were carried out at the University of South Bohemia (Ceske Budovice,
Czech Republic). These experiments included all samples from all profiles collected in each of three
study areas (CryoCarb 1-Kolyma in NE Siberia; CryoCarb 2-Taymyr in N Siberia; and CryoCarb 3-
Seida in NE European Russia). In total c. 1000 samples were incubated.
The Dry Bulk Density (DBD) of samples used for incubation was measured at Stockholm
University (Sweden). The %C and %N of dry weight were measured in an EA 1110 Elemental
Analyzer (CE Instruments, Milan, Italy) at Stockholm University (Seida samples) and the University
of Vienna (Kolyma and Taymyr samples).
Collected soil samples were dried at 40-50 °C within two weeks after field sampling and kept in
a cold room until analyzed. Dry soil (0.2 g) was mixed with 1.6 ml of soil inoculum (soil:$H_2O$,
1:100, weight/volume) in 10 ml vacutainers, after which the vacutainers were hermetically closed
and the soil slurry was incubated in an orbital shaker at 12 °C for 96 hours. At the end of incubation,
$CO_2$ concentration in the headspace was analyzed using an HP 5890 gas chromatograph (Hewlett-



Packard, USA), equipped with a TC detector. The soil inoculi were prepared from samples of fresh
soil taken separately from topsoil organic layer, mineral/peat active layer and mineral/peat
permafrost layer, from soil profiles collected in each study area. The fresh soil was kept in a cold
room and then conditioned at 15 °C for one week before inoculum preparation. Samples were
incubated with inoculum prepared from the respective soil horizons.
In the CryoCarb experiments, short term C-$CO_2$ production rates after rewetting of dried soil
samples was used as an indicator of SOM lability. It is well documented that C is released after
rewetting of dry soil, the amount of which is site and soil type specific and represents the available
fraction of soil C (Fierer and Schimel, 2003; Franzluebbers et al., 2000; Šantrůčková et al., 2006).
Due to different sample pretreatment, including duration until drying and incubation experiment, as
well as the different 'local' soil inoculi used, we consider the CryoCarb incubations of the three
different study areas as separate experiments.

*2.4. Geochemical parameters and C-$CO_2$ production rates*

As potential explanatory geochemical parameters we have considered dry bulk density (DBD),
carbon content (%C of dry weight) and carbon to nitrogen weight ratios (C/N). In this study, we
focus on the relationship between %C in samples and the corresponding C-$CO_2$ production in aerobic
incubation experiments. An important practical reason is that %C is most widely available since it
can be derived with a high degree of confidence from Elemental Analysis, but also from indirect
methods such as loss-on-ignition at 550 °C. However, there are also theoretical considerations for the
choice of %C. DBD is expected to be related to quantity and degree of compaction (decomposition)
of SOM. However, in the permafrost layer of soils it will also co-vary with the volume of excess
ground ice. C/N is a good indicator of degree of SOM decomposition in peat deposits (Kuhry and
Vitt, 1996) and tundra upland soils (Ping et al., 2008). Recent soil carbon inventories in permafrost
terrain have shown a clear decrease in soil C/N as a function of age/depth (e.g., Hugelius et al., 2010;
Palmtag et al., 2015). However, C/N is also sensitive to original botanical composition of the
peat/soil litter. In contrast, the %C of plant material is much more narrowly constrained to around 50
% of dry plant matter. For instance, based on data in Vardy et al. (2000), we can calculate a C/N
range of 48.5 ± 27.9 (mean and standard deviation) in modern phytomass samples from permafrost
peatlands in the Canadian Arctic (n=27) that included vascular plants, mosses and lichens. The
corresponding %C range was much narrower at 47.3 ± 5.1. An additional benefit of using %C is that
it has a clear 'zero' intercept in regressions against C-$CO_2$ production per gram dry weight per day
(i.e., at zero %C in soil samples we can expect zero C release). This is also the reason why
expressing C release as a function of gram dry weight (gdw) is more straightforward than against
gram C (gC). The latter would have the benefit of expressing C release directly as a function of C
stock, but the relationship is complex with recent studies showing high initial C release rates per gC
at low %C values (Weiss et al., 2016; Faucherre et al., 2018). In this study DBD is available for all
samples, and we can also express C release as a function of soil volume ($cm^3$). In the results we
primarily show µgC-$CO_2$ production per gdw per day (µgC-$CO_2$ $gdw^{-1}$ $d^{-1}$) as a function of %C in
the sample. However, in the Supplementary Materials we also refer to regressions against C/N and
C-$CO_2$ production rates per gC per day (µgC-$CO_2$ $gC^{-1}$ $d^{-1}$) or per $cm^3$ of soil per day (µgC-$CO_2$ $cm^{3-1}$
$d^{-1}$) against %C to test the robustness of our results.

*2.5. Landscape partitioning*

We have investigated C-$CO_2$ production rates for the full datasets as well as for samples grouped into
landscape unit classes that can be used for an assessment of vulnerable C pools at northern





circumpolar levels. For this purpose, we have subdivided our datasets to reflect the main Gelisol
suborders, non-permafrost mineral soils and Histosols recognized in the spatial layers of the
NCSCD, as well as deeper Quaternary deposits for which there are separate estimates of spatial
extent, depth and SOC stocks (Tarnocai et al., 2009; Strauss et al.,2013; Hugelius et al., 2014). We
identify the following landscape classes: peat deposits (Histels, and some Histosols), peaty wetland
deposits (mostly Histic Gelisols), mineral soils (Turbels and Orthels, and some non-permafrost
mineral soils), fluvial/deltaic (alluvial) deposits, and eolian/Yedoma deposits. Special attention is
paid to the lability of SOM in Holocene peat deposits, in deeper C-enriched buried layers and
cryoturbated pockets, and in Pleistocene Yedoma deposits. All classes are represented in the
PAGE21 and CryoCarb 1-Kolyma incubation experiments. The CryoCarb 2-Taymyr dataset lacks
sites with eolian parent materials, whereas the CryoCarb 3-Seida dataset does not include soils
formed into either alluvial or eolian deposits. We, therefore, focus on results from the PAGE21 and
CryoCarb 1-Kolyma experiments but present the main results from the two other experiments in the
Supplementary Materials.
*2.6. Statistics*
Relationships between $C-CO_2$ production rates and geochemical parameters for all samples, as well
as for groupings of samples into landscape unit classes, for each incubation experiment separately,
are statistically analyzed using linear, polynomial and other non-linear regressions in the Microsoft
Excel 2010 and Past3 (Hammer et al., 2001) software packages. Regressions are considered
significant if $p<0.05$. These analyses visualize SOM lability for full profiles including samples from
topsoil organic to mineral layers that have a wide range of DBD, %C and C/N values. In some cases,
replicates are not normally distributed (or even unimodal) and statistics should be interpreted with
caution. This is particularly the case in peatland profiles, with clusters of samples with low DBD and
high %C and C/N in the peat and opposite trends in samples of the underlying mineral subsoil.
To alleviate the issue on non-normal distributions, $C-CO_2$ production rates in samples as a
function of %C are also tested grouped into soil horizons. This approach yields classes that are much
better constrained in terms of %C values. Because data were still not fully normally distributed, non-
parametrical Mann-Whitney tests were used (Hammer et al., 2001). The data were log-transformed
to reduce skewness in data distributions and to reduce the influence of fractional data. For this
approach, we express $C-CO_2$ production rates per gC to take into account the large differences in %C
among the different soil horizon classes. The tests are run to evaluate null hypotheses regarding
differences in SOM lability between soil horizon classes, with a focus on those that are considered
typical for specific landscape classes (C-enriched pockets for Turbels, peat samples for Histels and
loess samples for Pleistocene Yedoma).
3. Results
*3.1. Simple geochemical indicators of SOM lability*
We first assessed the relationship between C release rates in incubation experiments and widely
available physico-chemical parameters in samples from soil carbon inventories carried out
throughout the northern permafrost region. The latter include dry bulk density (DBD), C content as a
percentage of dry sample weight (%C), and carbon to nitrogen weight ratios (C/N). In recent studies
dealing with incubation of soil samples from the northern permafrost region, %C and C/N of soil
samples were highlighted as best parameters to predict C release (Elberling et al., 2013; Schädel et
al., 2013). DBD was highlighted as a useful proxy in the recent synthesis of PAGE21 incubation
studies presented in Faucherre et al. (2018). All three parameters are significantly (anti-)correlated
with each other in the four different incubation experiments (Table 1 and Fig. S1). This can be
expected, since organically enriched topsoil samples have low DBD, high %C and high C/N values
compared to mineral layer soil samples. Also deeper soil samples, C-enriched through the process of
cryoturbation (Bockheim, 2007), have generally relatively low DBD, high %C and high C/N values
compared to adjacent mineral soil samples (e.g. Hugelius et al., 2010; Palmtag et al., 2015).
Table 1. $R^2$ values of cross correlations between three geochemical parameters for all samples in the
PAGE21 and three CryoCarb incubation experiments (all significant, $p < 0.05$). For regression
models see Fig. S1.

All three considered geochemical parameters are significantly (anti-)correlated with measured C
release rates in the four different incubation experiments. Lower DBD, higher %C and higher C/N
values are associated with higher C-$CO_2$ production per gdw of the samples (Table 2 and Fig. S2).
Of the three parameters, DBD explains most of the observed variability in C release in two
experiments, whereas C/N shows highest $R^2$ values in the other two experiments.
Table 2. $R^2$ values of regressions between three geochemical parameters and µgC-$CO_2$ production
304        per gram dry weight for all samples in the PAGE21 and the three CryoCarb incubation
experiments (all significant, $p < 0.05$). For regression models see Fig. S2.

*3.2. Partitioning of the datasets based on landscape unit classes*
Our results show a significant relationship between µgC-$CO_2$ production per gdw as a function of
%C of the soil sample for the full datasets in each of the four incubation experiments (Fig. S2).
However, less than 50 % of the variability is explained by this relationship, which implies that it has
limited usefulness to predict C release based on %C of the samples only (Table 2). In this section we
analyze whether a grouping of samples according to landscape unit classes can disentangle some of
the observed variability.
Figure 2 shows the significant relationships between C release rates and %C in the samples for
the full datasets in the PAGE21 (measured on day 343 of incubation) and CryoCarb 1-Kolyma
(measured over the first four days of incubation) experiments. A first observation is that C release
rates per gdw are c. 15 times lower in the longer-term PAGE21 experiment compared to the short-
term CryoCarb 1-Kolyma experiment. Both experiments show a large range in C release, particularly
at medium to high %C values. Figure 3 shows the same two experiments and data points, but
grouped according to major landscape unit classes. For the sake of simplicity, we apply linear
regressions with intercept zero to all groups. The linear regression for the full data set is provided as
reference, but it should be noted that its slope is partly determined by the number of samples in each
of the recognized landscape units. These are identified by different colors and symbols that have
been consistently applied in Figs. 3-4 and S3-S5.
Figure 2. µgC-$CO_2$ production per gram dry weight as a function of %C of the sample for the full
datasets in the (a) PAGE21 (top panel) and (b) CryoCarb 1-Kolyma (lower panel) incubation
experiments (both regressions significant, $p < 0.05$).

In the PAGE21 dataset (Fig. 3a), the soils developed into alluvial and eolian deposits and in
peaty wetlands all show similar and relatively high SOM lability. Mineral soils show intermediate





values, whereas the peat deposits display low SOM lability (when considering %C values). All
regressions are significant, except for 'peat deposits' due to very high variability in three surface peat
samples (but see Fig. 4d). In the CryoCarb 1-Kolyma data set (Fig. 3b), alluvial and eolian
soils/deposits show the highest SOM lability, followed by mineral soils. In this case, peaty wetlands
show a slightly lower lability than mineral soils/deposits but still considerably higher than peatlands.
This clear dichotomy in the SOM lability of mineral soils (including peaty wetlands) and peat
deposits is also apparent from the CryoCarb 2-Taymyr and CryoCarb 3-Seida results even though not
all landscape classes are represented in those experiments (Fig. S3). The explanatory power of the
regressions ($R^2$ values) in the peatland class is generally lower than that in the mineral soil/deposit
classes. These statistics are, however, greatly improved when removing the surface peat samples
from the analyses (not shown), which display very high variability.
Figure 3. µgC-$CO_2$ production per gram dry weight as a function of %C of the sample for the
different landscape classes in the longer-term PAGE21 (a, top panel) and short-term CryoCarb
1-Kolyma (b, lower panel) incubation experiments: Alluvial class (red line and diamonds);
Eolian class (blue line and triangles); Mineral class (brown line and squares); Peaty wetland
class (dark green line and circles); Peatland class (light green line and circles). All regressions
significant, p<0.05, except for the PAGE21 peatland class (n.s.).

Linear regression analyses between C-$CO_2$ production per gdw and C/N ratios for all four
experiments (Fig. S4) show small deviations from the above patterns but generally maintain the clear
difference between 'peat deposits' and the remaining landscape units. However, peat deposits with
low C/N values (≤20) seem to decompose at similar rates as SOM in mineral soils and deposits with
similar C/N ratios. $R^2$ values for the landscape classes are generally lower than in regressions against
%C and regression lines at low C release tend to converge to C/N values of 8-12, which are typical
for microbial decomposer biomass suggesting only slow internal cycling of remaining SOM
(Zechmeister-Boltenstern et al., 2015).
The PAGE21 dataset with C-$CO_2$ production rates expressed per gC as a function of %C of the
soil sample also shows similar results, however, with generally lower $R^2$ and sometimes non-
significant regressions (Fig. S5a). The same patterns are also noted when expressing C release as a
function of soil volume ($cm^3$), however, $R^2$ values are generally even lower and more often non-
significant (Fig. S5b).
*3.3. Further subdivision of landscape unit classes in the PAGE21 dataset*
In Fig. 4, landscape unit classes in the PAGE21 dataset have been further subdivided and different
functions (second order polynomial or exponential) providing better fits have been applied. The
eolian class is subdivided into actively accumulating deposits (Adventdalen) and Holocene soils
formed into Pleistocene Yedoma parent materials (Lena Delta), and specifically identifies buried C-
enriched samples (Fig. 4a). Alluvial deposits are separated into profiles from active and pre-recent
floodplains (multiple study areas), again separating samples from deeper C-enriched buried layers
and cryoturbated pockets (Fig.4b). Mineral soils are separated into active colluviation sheets
(mountain slopes on Svalbard) and other mineral soils (multiple study areas), highlighting the one
buried C-enriched sample found in this class (Fig. 4c). Generally speaking, a second order
polynomial (intercept zero) provides the best fit and has been applied for the sake of uniformity to all
described subclasses. All these three datasets have in common that the subclasses with active surface
accumulation/movement have topsoil samples that show relatively low C content due to the
continuous admixture of minerogenic materials. At the same time, these all show the highest C-$CO_2$



production per gdw (when considering %C). Furthermore, the second order polynomial regressions
of all subclasses (except for buried C-enriched samples) suggest that the topsoil samples are
particularly labile suggesting the presence of a 'fast' SOM pool in the recently deposited plant litter.
Deeper C-enriched material shows relatively low lability and does not show rapidly increasing
lability at higher %C values.

Figure 4d compares the SOM lability in peat deposits (fens and bogs in Stordalen Mire) and
peaty wetland profiles (multiple study areas), adding for comparison the results from the previously
described deeper C-enriched buried layers and cryoturbated pockets in mineral soils (see Figs. 4a-c).
In this case, exponential functions best describe observed trends, pointing to very high lability of
surface peat(y) samples. The thin peat layers in peaty wetlands have relatively low %C values
pointing to admixture of minerogenic materials. The SOM in these profiles show relatively high C-
$CO_2$ production per gdw compared to 'true' peat samples (when considering %C). Compared to the
non-significant linear regression for all peat samples shown in Fig. 3a, exponential regressions for
the peatland class as a whole as well as for fens and bogs separately are statistically significant.
Particularly in fen peat, this regression is able to capture some very high C release values of two
surface peat samples (corresponding to graminoid-derived plant litter). Deeper C-enriched material
in mineral soils displays only slightly higher SOM lability compared to the mineral subsoil
underlying peat deposits. It is important to bear in mind that the total number of peat samples from
Stordalen Mire is limited (n=13) and that results cannot be compared directly to adjacent mineral soil
profiles because field sampling in that particular study area focused solely on the peatland area.
Figure 4. µgC-$CO_2$ production per gram dry weight as a function of %C of the sample for different
landscape classes and their subdivisions in the PAGE21 incubation experiment. (a) Eolian class
separated into actively accumulating deposit (light blue), Holocene soil formation into
Pleistocene Yedoma parent materials (dark blue) and buried C-enriched samples (pink); (b)
Alluvial class separated into active floodplain (rose), Holocene soil formation into pre-recent
floodplain deposits (red) and buried C-enriched samples (pink); (c) Mineral class separated into
active colluviation sheet (light brown), other mineral soils (dark brown) and buried C-enriched
samples (pink); (d) wetland class separated into wetlands with thin peat layers (green), fens
(light green) and bogs (dark green) with deep peat deposits and, for comparison, buried C-
enriched samples in mineral soils (pink). The hatched line represents the regression for all true
peatland samples (fens and bogs) together. C-release from one surface peat sample (green-
orange) in the margin of a peatland is also indicated, but not included in the regressions. All
regressions are significant ($p<0.05$).

*3.4. C-enriched cryoturbated and Pleistocene Yedoma samples in the CryoCarb 1-Kolyma dataset*
In the PAGE21 incubation each profile included only one sample from the mineral soil in the middle
of the active layer and one sample from the upper permafrost layer. Thus, the selection of samples
was based on depth-specific criteria. As a result, the number of samples from deeper C-enriched
buried layers and cryoturbated pockets is limited (n=13). In the CryoCarb 1-Kolyma experiment
samples from entire profiles were incubated and the number of deeper C-enriched samples in the
mineral soil horizons is much larger. Figure 5a compares the C-$CO_2$ production per gdw from
organically-enriched topsoil and mineral soil samples not affected by C-enrichment with that in
deeper C-enriched cryoturbated samples in tundra upland profiles. For the sake of clarity, only those
cryoturbated samples which are C-enriched by at least twice the adjacent mineral soil %C
background values are included (n=22). The results from this much more narrowly defined dataset
are similar to those presented for the PAGE21 experiment, i.e. SOM in deeper C-enriched
cryoturbated samples is less labile than in organically-enriched topsoil samples with similar %C.



The PAGE21 experiment does not include any samples from Pleistocene Yedoma deposits. In
contrast, the CryoCarb 1-Kolyma dataset includes samples from two Yedoma exposures along river
and thermokarst lake margins. The material was collected from perennially frozen Yedoma deposit
as well as from thawed out sections of the exposures. C-release from these samples are presented in
Fig. 5b, which for comparison also shows low %C samples from the upper permafrost horizon in
Holocene tundra soils formed into Yedoma parent materials, mineral subsoil samples beneath peat
deposits and samples from deeper C-enriched cryoturbated samples. The C-$CO_2$ production per gdw
of Pleistocene Yedoma is lower than that of permafrost horizon samples in Holocene soils, but
somewhat higher to that of samples from mineral subsoil beneath peat and deeper C-enriched
samples (when considering %C). Furthermore, the SOM lability of thawed out deposits is somewhat
lower than that of the intact permafrost Yedoma material.

Figure 5. µgC-$CO_2$ production per gram dry weight as a function of %C of the sample in the
CryoCarb 1-Kolyma incubation experiment for (a) deeper C-enriched samples (pink line and
triangles), compared to organically enriched topsoil and mineral soil samples not affected by C-
enrichment in Holocene tundra upland profiles (blue line and triangles), and for (b) perennially
frozen Pleistocene Yedoma samples (black line and triangles) and thawed out Pleistocene
Yedoma samples (red line and triangles), compared to upper permafrost layer samples in
Holocene tundra upland soils (blue line and triangles), mineral subsoil samples beneath peat
deposits (green line and circles, showing start of regression line) and buried C-enriched samples
(pink line and triangles, showing start of regression line). All regressions (power fit) are
significant ($p < 0.05$).

*3.5. Relative lability ranking of SOM landscape unit classes*

Table 3a shows the slopes of the linear regressions (intercept zero) between C-$CO_2$ production per
gdw and %C of samples for the different landscape unit classes in all four incubation experiments.
From these results it is clear that results from the four experiments cannot be compared directly in
quantitative terms. To facilitate comparison across experiments the results were therefore normalized
to the lowland mineral soil class, which consistently showed intermediate SOM labilities. Table 3b
shows the normalized regression slopes (with the slope for mineral soils set to 1), and their mean and
standard deviation (when the landscape class is represented in more than one incubation experiment).
This approach confirms the previous results that peat deposits and deeper C-enriched samples in
mineral soils consistently show very low relative lability, whereas areas with recent mineral sediment
accumulation (in active floodplains and recent eolian deposits) display generally somewhat higher
SOM lability (when considering %C). Pleistocene Yedoma deposits, only represented in one
incubation experiment, also display relative low SOM lability.

Table 3. (a) Slopes of linear regressions (intercept zero) between %C and C-$CO_2$ production per gdw
in samples of the different landscape classes in the four experiments; (b) Normalized slopes of
linear regressions between %C and C-$CO_2$ production per gdw for samples in the different
landscape classes in the four experiments (slope of mineral soils in lowland settings set to 1).

*3.6. SOM lability based on soil horizon criteria*

We also tested SOM lability in samples grouped according to soil horizon criteria, with special
attention to those horizon classes that can be linked to the specific landscape units that show low
relative SOM lability (C-enriched pockets for Turbels, peat samples for Histels and loess samples for





Pleistocene Yedoma). This approach yielded classes with data distributions that are much better
constrained in terms of %C values.
For the mineral soils in the PAGE21 incubation experiment, we differentiated between the
topsoil organic layer, the active layer mineral soil, the permafrost layer mineral soil, and C-enriched
pockets in both active layer and permafrost layer. Samples from topsoil organic layer, the active
layer mineral soil and the permafrost layer mineral soil from profiles formed in Late Holocene loess
deposits in Adventdalen (Svalbard) are considered separately, as are the active layer peat samples
from Stordalen Mire (N Sweden). A similar grouping has been made for mineral soils in the
CryoCarb 1-Kolyma experiment. In this case, Pleistocene Yedoma loess samples (both frozen and
thawed) are considered separately. Peat samples are much better represented in the CryoCarb 1-
Kolyma than PAGE21 experiment, and are subdivided into samples from thin peat layers in the
active layer of peaty wetlands (Histic Gelisols), as well as samples from the active layer and
permafrost layer of deep peat deposits (Histels).
In this analysis we focus on C-$CO_2$ production per gC to take into account large differences in
%C between soil horizon classes (see Fig. S6). The main difference between the two experiments is
the much lower %C values of the topsoil organic class in the PAGE21 incubation, which can be
explained by a greater surface admixture of minerogenic material in alluvial (Lena Delta), eolian and
mountainous areas (Svalbard). In contrast, the predominant lowland setting of the CryoCarb 1-
Kolyma study area is characterized by soils with thicker, more C-rich, topsoil organic layers.
Figure 6 shows C-$CO_2$ production per gC in the soil horizon groups of the longer term PAGE21
and short-term CryoCarb 1-Kolyma experiments. Results of the Mann-Whitney paired tests for both
these experiments are shown in Table 4. PAGE21 classes show fewer statistically significant
differences than in the CryoCarb 1-Kolyma experiment, which can at least partly be ascribed to
smaller sample sizes. The number of samples in the PAGE21 incubation for C-enriched pockets in
the active layer (n=3) and for peat (n=6) are particularly low.

Figure 6. µgC-$CO_2$ production per gram carbon in samples of (a) the PAGE21 and (b) the CryoCarb
1-Kolyma incubation experiments, grouped according to soil horizon criteria. Abbreviations:
AL-OL = Active layer topsoil organics; AL-Min = Active layer mineral; AL-Ce = Active layer
C-enriched; P-Min = Permafrost layer mineral; P-Ce = Permafrost layer C-enriched; AL-Pty =
Active layer thin peat (CryoCarb 1-Kolyma experiment only); AL-Pt = Active layer peat; P-Pt =
Permafrost layer peat (CryoCarb 1-Kolyma experiment only); AL-Lss OL = Active layer topsoil
organics in Late Holocene loess deposits (PAGE21 experiment only); AL-Lss Min = Active
layer mineral in Late Holocene loess deposits (PAGE21 experiment only); P-Lss Min =
Permafrost layer mineral in Late Holocene loess deposits (PAGE21 experiment only); P-Yed =
Permafrost Pleistocene Yedoma deposits (CryoCarb 1-Kolyma experiment only); Th-Yed =
Thawed out Pleistocene Yedoma deposits (CryoCarb 1-Kolyma experiment only). Box-whisker
plots show mean and standard deviation (in red) and median, first and third quartiles and
min/max values (in black), for the different soil horizon groups.

Table 4. p Values of Mann-Whitney paired tests of µgC-$CO_2$ production per gram carbon for soil
horizon groups in (a) the PAGE21 and (b) the CryoCarb 1-Kolyma incubation experiments.
Abbreviations as in Fig. 6. Differences are considered significant when p<0.05.

The C release rates in topsoil organic samples from the actively accumulating Holocene loess
soils are significantly higher than those in topsoil organic samples from the remaining PAGE21
mineral soils (Fig. 6a and Table 4a). Both topsoil organic classes show significantly higher rates than
all mineral soil and peat classes. Peat samples have the lowest mean and median C release rates from





all these classes but only the rates from permafrost layer mineral soil and C-enriched pocket samples
are significantly higher. Both mean and median C release rates from active layer and permafrost
layer C-enriched pockets are somewhat lower (but not significantly different) than those from
adjacent, non C-enriched, mineral soil samples.
C release rates in the soil horizon classes from the CryoCarb 1-Kolyma experiment show
similarities, but also some differences to those observed in the PAGE21 experiment. Absolute C
release rates per gC are more than an order of magnitude higher in the CryoCarb 1-Kolyma
experiment (measured as a mean release over the first four days of incubation) compared to those in
the PAGE21 experiment (measured at day 343). Another important difference is that C release rates
per gC in the short-term CryoCarb 1-Kolyma incubation do not differ significantly between the
topsoil organic class and the active layer and permafrost layer mineral soil classes, which we ascribe
to the presence of a highly labile C pool (e.g. DOC, plant roots) in the mineral soil layers that is
quickly decomposed (see Weiss et al., 2016; Faucherre et al., 2018). However, rates from active
layer and permafrost layer C-enriched pockets are significantly lower than those from adjacent, non
C-enriched, mineral soil samples. Both active layer and permafrost layer peat samples show
significantly lower C release rates than all other classes, with active layer peat samples having
significantly higher rates than permafrost layer peat samples. Samples from the Pleistocene Yedoma
loess 'frozen' and 'thawed' classes display significantly lower C release rates per gC than those in
the topsoil organic layer, active layer and permafrost layer mineral soil classes, but significantly
higher rates than those in the peat classes. The two Yedoma classes do not differ significanty from
each other, the active layer and permafrost layer C-enriched pocket classes, nor the peaty wetland
class.

4. Discussion

The analysis and comparison of results in the PAGE21 and CryoCarb 1-Kolyma incubations show
consistent trends in C-$CO_2$ production rates as a function of simple soil geochemical parameters in
both the full datasets as well as in the grouping of samples according to landscape classes. However,
it is not possible to directly compare these two very different laboratory experiments quantitatively.
The varying field collection techniques, field storage, transport and laboratory storage, pretreatment,
experimental setup and time of measurement after start of incubations have a clear effect on the
magnitude of the observed C-$CO_2$ production rates. The same methods were applied to all samples
from all study areas in the PAGE21 experiment, but these differed markedly from those applied in
the CryoCarb setup and even between the three individual CryoCarb experiments (e.g., addition of
different 'local' microbial decomposer inoculi to rewetted samples).
In quantitative terms, C-$CO_2$ production rates per gdw measured over the first 4 days in the
CryoCarb 1-Kolyma samples incubated at 12 ºC are about 15 times higher than those after about one
year in the PAGE21 samples incubated at 5 ºC (see Fig. 2). Similarly, C-$CO_2$ production rates per gC
are also more than an order of magnitude higher in the short-term CryoCarb-Kolyma 1 than the
longer term PAGE21 incubation (see Fig. 6). Upper permafrost mineral soil samples (<3 %C) from
Kylatyk in NE Siberia, incubated at 2 ºC directly after field collection and thawing (measurement
after 20-30 hr, following 3 days of pre-incubation), show median C release rates of c. 750 µgC-$CO_2$
$gC^{-1}$ $d^{-1}$ (Weiss et al., 2016), compared to c. 1750 µgC-$CO_2$ $gC^{-1}$ $d^{-1}$ in the same class of CryoCarb 1-
Kolyma samples. Median C release rates in upper permafrost mineral soil samples of the PAGE21
experiment (Faucherre et al., 2018) decrease from c. 170 on day 8 to c. 35 µgC-$CO_2$ $gC^{-1}$ $d^{-1}$ on day
343 since start of incubation. It is obvious from these results that there is a rapid decline in C release
rates over time of incubation. Longer incubation experiments (up to 12 years) have shown that the
overall rate of C loss decreases almost exponentially over time (Elberling et al., 2013). However,
even when laboratory incubation setups and time of measurement are similar, large differences can





occur in C release rates. For instance, peat samples in the CryoCarb 1-Kolyma incubation display
about twice the C-$CO_2$ production rates per gdw than those observed in the CryoCarb 3-Seida
incubation (Figs. 3b and S3b).
Nonetheless, a comparison of C-$CO_2$ production rates per gdw for landscape unit classes in
terms of relative SOM lability provided useful and robust results. These classes were implemented to
allow upscaling of results to the northern permafrost region. They reflect main Gelisol (and non-
Gelisol) soil suborders and deeper Quaternary deposits to permit direct comparison with the size and
geographic distribution of these different SOC pools (Tarnocai et al., 2009; Hugelius et al., 2014).
Samples from mineral soil profiles, including wetland deposits with a thin peat(y) surface layer,
display high relative SOM lability compared to peat deposits, deep C-enriched buried or cryoturbated
samples and Pleistocene Yedoma deposits (when considering %C of the incubated sample). These
results are confirmed by the more stringent statistical analysis of samples grouped into soil horizon
classes. Peat deposit, C-enriched pocket and Yedoma deposit samples show significantly lower C-
$CO_2$ production rates per gC than topsoil organics and mineral layer samples (Cryo-Carb-1-Kolyma
experiment). The same trends are observed in the incubation experiment of upper permafrost samples
from Kytalyk, reported by Weiss et al. (2016). C-enriched pockets (3-10 %C) showed lower C-$CO_2$
production rates per gC than mineral soil samples (<3 %C), while a buried peat sample (c. 40 %C)
displayed a very low C-$CO_2$ production rate per gC. The PAGE21 experiment also revealed that peat
samples mineralized a smaller fraction of C over the one year of incubation compared to mineral soil
samples (Faucherre et al., 2018).
A further subdivision of landscape classes and more careful analysis of incubation results in the
PAGE21 experiment provide additional useful insights. For example, separation of eolian deposits
into actively accumulating deposits during the Late Holocene (Adventdalen) and Holocene soils
formed into Pleistocene Yedoma parent materials (Lena Delta) showed clear differences in C release
rates per gdw (when considering %C), with the former displaying a higher SOM lability in topsoil
organic samples (see Fig. 4a). The topsoil organic samples from the actively accumulating eolian
deposits in Adventdalen also displayed significantly higher C release rates per gC than all other
topsoil organic, mineral layer and peat(y) horizon classes (see Table 4a). Separation of alluvial
deposits into active floodplain deposits and Holocene soils formed in pre-recent river terraces and of
mineral soils into active colluviation sheets (mountain slopes on Svalbard) and other mineral soils
(multiple study areas) showed similar trends in SOM lability (see Fig. 4b-c). These results suggest
that admixture of minerogenic material in topsoil organics of actively accumulating eolian, alluvial
and colluvial deposits promotes SOM decomposition. Peaty wetland deposits display much higher C
release rates per gdw (when considering %C) than peatland deposits (see Fig. 4d). These two
landscape classes are poorly represented in the PAGE21 experiment, but a statistical test of C release
rates per gC in these peat(y) soil horizon classes of the CryoCarb 1-Kolyma incubation confirms this
difference (see Table 4b). This is interesting because even though wetlands with a thin peat layer do
not have particularly high C stocks, they can be important sources of methane ($CH_4$) to the
atmosphere (Olefeldt et al., 2013). These further subdivisions into landscape subclasses are of
limited use for upscaling purposes because they are not considered explicitly in any available
geographic database for the northern permafrost region.
The relatively low lability in the peatland class is surprising. The low DBD, high %C and high
C/N of peat are normally associated with a relatively low degree of decomposition. This, in turn, is
the result of environmental factors such as anaerobic and/or permafrost conditions that largely inhibit
SOM decay (Davidson and Janssens, 2006). One could expect that this less decomposed material
would show high lability following thawing and warming, but our results point to the opposite. This
is particularly surprising when considering the setup of the CryoCarb experiments, in which a slush
of rewetted material inoculated with microbial decomposers was incubated at 12 °C. The CryoCarb
experiments are very short assays (4 days), but the longer term PAGE21 experiment (measured after



roughly one year) provides similar results. As previously suggested by Capek et al. (2015), also
SOM from deeper C-enriched buried layers and cryoturbated pockets show relatively low lability
when compared to organically-enriched topsoil and mineral layer samples not affected by C-
enrichment in all incubation experiments. To these two categories of samples can be added
Pleistocene Yedoma deposits, which despite incorporation of relative fresh plant root material caused
by syngenetic permafrost aggradation, also display low relative SOM lability. This implies that SOM
in three of the major SOC pools in the northern permafrost region, i.e. deeper peat deposits in
Histels/Histosols, deeper C-enriched material in Turbels and Pleistocene Yedoma deposits, display a
high level of resistance to decomposition. The reason why this relatively undecomposed material
displays low lability remains unclear. One reason could be that the decomposer community needs
time to adapt to the new environmental conditions following thawing/warming, another one that
there is a simple mismatch between the microbial community adapted to decompose relatively
undecomposed organic material and the physico-chemical environment (e.g., higher bulk density)
prevailing in (thawed out) deeper soil horizons (Gittel et al., 2013; Schnecker et al., 2014). In the
case of peat deposits, it should also be considered if this resistance to decomposition is an evolved
'biochemical trait' in peat-forming species that maintains their favored habitat, similar to the role of
*Sphagnum* anatomy (hyaline cells), physiology (acidification) and cell wall chemistry (phenolic
compounds) in sustaining moist and acid surface conditions, and inhibiting peat decomposition
(Clymo and Hayward, 1982). Furthermore, the generally high C/N ratios of peat provide a poor
substrate quality to the decomposer community (Bader et al., 2018).
An important consideration is if the consistent differences in relative SOM lability of landscape
and soil horizon classes observed in our incubation experiments will be maintained over periods of
decades to centuries of projected warming and thawing. Very short-term incubations, such as in the
CryoCarb setup (four days), might register the initial decomposition of highly labile SOM
components, such as microbial necromass, simple molecules (e.g., sugars or amino acids), low
molecular-weight DOC, etc., or might not provide enough time for an adaptation of the microbial
decomposer community to new environmental settings (Weiss et al., 2016; Weiss and Kaal, 2018).
On the other hand, in longer incubation experiments such as in the PAGE21 experiment (one year),
the conditions in the incubated samples become gradually more artificial compared to field
conditions. Specifically, microbes in long-term incubations become increasingly C limited, as no
new C input by plants occur, whereas inorganic nutrients, such as nitrate or ammonium accumulate
to unphysiological levels. Care, therefore, should be taken when extrapolating our results over longer
time frames if no corroborating field evidence for longer term decay rates can be obtained (e.g.
Kuhry and Vitt, 1996; Schuur et al., 2009).

5. Conclusions

The PAGE21 and CryoCarb incubation experiments confirm results from previous studies that
simple geochemical parameters such as DBD, %C and C/N can provide a good indication of SOM
lability in soils and deposits of the northern permafrost region (Elberling et al., 2013; Schädel et al.,
2013; Faucherre et al., 2018.). In our analyses we have focused on %C of the sample since it is the
most widely available of the three investigated geochemical parameters. Furthermore, %C is less
sensitive than C/N to botanical origin of the plant litter and, in contrast to DBD, not dependent on
ground compaction or volume of excess ground ice.
When considering the full datasets of the four experiments, our regressions of C release as a
function of %C were statistically significant but explained less than 50 % of the observed variability.
Subsequently, we investigated whether a further division of samples into predefined landscape unit
classes would better constrain the observed relationships. In defining these classes, we applied a
scheme that could easily be used for spatial upscaling to northern circumpolar levels. We adopted the

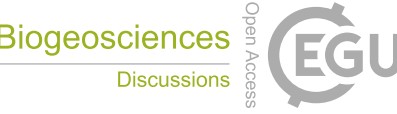

main Gelisol suborders (Histels, Turbels and Orthels), non-permafrost Histosols and mineral soils,
and types of deeper Quaternary (deltaic/floodplain and eolian/Yedoma) deposits used in the NCSCD
and related products to estimate the total SOC pool in the northern permafrost region (Tarnocai et al.,
2009; Strauss et al., 2013; Hugelius et al., 2014). We conclude that these landscape classes better
constrain observed variability in the relationships and that the relative SOM lability rankings of these
classes were consistent among all four incubation experiments, for both regressions against %C and
C/N (all four experiments), and for regressions of %C against different units of $C-CO_2$ production
*'per gram dry weight', 'per gram C' and 'per cm$^3$'* (PAGE21 dataset). Our results based on full
profiles indicate that $C-CO_2$ production rates per gdw decrease in the order Late Holocene eolian >
alluvial and mineral upland (including peaty wetlands) > Pleistocene Yedoma > C-enriched pockets
> peat, with lowest C release rates observed in peat deposits (when considering %C). These results
are corroborated by statistical analysis of C release rates per gC for samples grouped according to
soil horizon criteria (PAGE21 and CryoCarb 1-Kolyma datasets).
An important conclusion from these results is that purportedly more undecomposed SOM, such
as in peat deposits (Histels and Histosols), C-enriched cryoturbated samples (Turbels), and
Pleistocene Yedoma deposits, does not seem to imply higher SOM lability. These three SOC pools
together represent ≥50 % of the reported SOC storage in the northern permafrost region (Hugelius et
al., 2014; Palmtag and Kuhry, 2018). Consequently, there is an urgent need for further research to
understand these results in order to better constrain the thawing permafrost carbon feedback on
global warming.

6. Data availability

The soil geochemical data and incubation results presented in this paper are available upon request
from PK (peter.kuhry@natgeo.su.se). For full PAGE21 incubation data, please contact BE
(be@ign.ku.dk). For full CryoCarb incubation data, please contact JB (jiri.barta@prf.jcu.cz).

7. Author contribution

PK developed the initial concept for the study. All authors contributed with the collection of soil
profiles at various sites. The PAGE21 incubation experiment was planned and conducted at
CENPERM (University of Copenhagen) by SF and BE, whereas the CryoCARB incubation
experiments were carried out at the University of South Bohemia (Ceske Budejovice) under
guidance of HS and JB. PK performed all statistical analyses, in cooperation with GH. All co-authors
contributed to the writing of the manuscript, including its discussion section.

8. Competing interests

The authors declare that they have no conflict of interest.

9. Acknowledgments

Collection and laboratory analyses for Svalbard (Adventdalen and Ny Ålesund), Stordalen Mire and
Lena Delta samples were supported by the EU-FP7 PAGE21 project (grant agreement no 282700).
Lower Kolyma and Taymyr Peninsula samples were collected and incubated in the framework of the
ESF-CryoCarb project, with support from the Swedish Research Council (VR to Kuhry), the



Austrian Science Fund (FWF I370-B17 to Richter), the Czech Science Foundation (Project 16-
18453S to Barta) and the Czech Soil-Water Research Infrastructure (MEYS CZ grants LM2015075
and EF16-013/0001782 to Šantrůčková). Seida samples were originally collected in the framework
of the EU-FP6 Carbo-North project (contract 036993). Gustaf Hugelius acknowledges a Swedish
Research Council Marie Skłodowska Curie International Career Grant. We thank Christian Jungner
Jørgensen (University of Copenhagen) for guidance and assistance in the PAGE21 incubation
experiment. Kateřina Diaková is acknowledged for the collection of the soil inoculi in Seida. The
Seida samples were subsequently incubated at the University of South Bohemia. We are most
grateful to Nikolai Lashchinskiy (Siberian Branch of the Russian Academy of Sciences,
Novosibirsk) and Nikolaos Lampiris, Juri Palmtag, Nathalie Pluchon, Justine Ramage, Matthias
Siewert and Martin Wik (all Stockholm University), for help in sample collection. We would also
like to thank Magarethe Watzka (University of Vienna) for elemental analyses of soil samples.
Zhanna Kuhrij is acknowledged for the preparation of Figs. 6 and S6.

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






Table 1

| Correlation | Number samples | %C vs C/N | %C vs DBD | C/N vs DBD |
|---|---|---|---|---|
| | | positive | negative | negative |
| PAGE21, All sites | 238 | 0.53 | 0.67 | 0.57 |
| CryoCarb 1-Kolyma | 442 | 0.79 | 0.78 | 0.63 |
| CryoCarb 2-Taymir | 502 | 0.64 | 0.69 | 0.47 |
| CryoCarb 3-Seida | 80 | 0.47 | 0.84 | 0.47 |






Table 2

| Correlation | Number samples | DBD vs C release/gdw negative | %C vs C release/gdw positive | C/N vs C release/gdw positive |
|---|---|---|---|---|
| PAGE21, All sites | 238 | 0.52 | 0.45 | 0.34 |
| CryoCarb 1-Kolyma | 442 | 0.52 | 0.47 | 0.54 |
| CryoCarb 2-Taymir | 502 | 0.41 | 0.33 | 0.48 |
| CryoCarb 3-Seida | 80 | 0.81 | 0.43 | 0.38 |





Table 3

a

| | Pt | Min/CE | Min Mtn | Min Pty | Min Lowl | Alluv | Eol | Pl Yed |
|---|---|---|---|---|---|---|---|---|
| PAGE21, All sites | 0.20 | 0.29 | 0.99 | 1.51 | 1.44 | 1.43 | 1.68 | |
| CryoCarb-Kolyma | 4.83 | 6.72 | 17.8 | 15.3 | 19.4 | 22.0 | | 11.5 |
| CryoCarb-Taymir | 6.24 | | | 29.3 | 24.7 | 26.2 | | |
| CryoCarb-Seida | 2,40 | | | 5.76 | 7.92 | | | |

b

| | Pt | Min/CE | Min Mtn | Min Pty | Min Lowl | Alluv | Eol | Pl Yed |
|---|---|---|---|---|---|---|---|---|
| PAGE21, All sites | 0.14 | 0.20 | 0.69 | 1.05 | 1 | 0.99 | 1.17 | |
| CryoCarb-Kolyma | 0.25 | 0.35 | 0.91 | 0.79 | 1 | 1.12 | | 0.59 |
| CryoCarb-Taymir | 0.26 | | | 1.18 | 1 | 1.06 | | |
| CryoCarb-Seida | 0.30 | | | 0.73 | 1 | | | |
| Mean relative lability | 0.24 | 0.28 | 0.80 | 0.94 | 1 | 1.06 | 1.17 | 0.59 |
| S.D. relative lability | 0.07 | 0.11 | 0.16 | 0.21 | | 0.07 | | |

Abbreviations: 'Pt' = peat deposits (Histels/Histosols); 'Min/CE' = C-enriched pockets in cryoturbated soils (Turbels); 'Min Mtn' = mineral soils in mountain settings; 'Min Pty' = peaty wetlands (mineral soils with histic horizon); 'Min Lowl' = mineral soils in lowland settings; 'Alluv' = recent alluvial deposits and Holocene soils formed in alluvial deposits; 'Eol' = recent eolian deposits and Holocene soils formed in eolian deposits; 'Pl Yed' = Pleistocene Yedoma deposits




Table 4

**a**

| | AL_Min | AL_Ce | P_Min | P_Ce | AL_Pt | AL_OL Ls | AL_Min Ls | P_Min Ls | |
|---|---|---|---|---|---|---|---|---|---|
| AL_OL | < 0.001 | 0.0072 | < 0.001 | < 0.001 | < 0.001 | 0.0493 | < 0.001 | < 0.001 | AL_OL |
| AL_Min | | 0.5761 | 0.2217 | 0.5360 | 0.1887 | < 0.001 | 0.6598 | 0.5682 | AL_Min |
| AL_Ce | | | 0.1464 | 0.1387 | 0.5186 | 0.0160 | 0.1956 | 0.7096 | AL_Ce |
| P_Min | | | | 0.5570 | 0.0119 | < 0.001 | 0.7353 | 0.0809 | P_Min |
| P_Ce | | | | | 0.0119 | < 0.001 | 1.0000 | 0.1828 | P_Ce |
| AL_Pt | | | | | | 0.0018 | 0.0518 | 0.1103 | AL_Pt |
| AL_OL Ls | | | | | | | < 0.001 | < 0.001 | AL_OL Ls |
| AL_Min Ls | | | | | | | | 0.2500 | AL_Min Ls |

**b**

| | AL_Min | AL_Ce | P_Min | P_Ce | AL_Pty | AL_Pt | P_Pt | Fr_Yed | Th_Yed | |
|---|---|---|---|---|---|---|---|---|---|---|
| AL_OL | 0.3800 | 0.0027 | 0.0658 | < 0.001 | 0.0255 | < 0.001 | < 0.001 | < 0.001 | < 0.001 | AL_OL |
| AL_Min | | < 0.001 | 0.011 | < 0.001 | 0.0174 | < 0.001 | < 0.001 | < 0.001 | < 0.001 | AL_Min |
| AL_Ce | | | 0.1178 | 0.2318 | 0.8849 | 0.0083 | < 0.001 | 0.3428 | 0.1653 | AL_Ce |
| P_Min | | | | < 0.001 | 0.1656 | < 0.001 | < 0.001 | 0.0017 | < 0.001 | P_Min |
| P_Ce | | | | | 0.4539 | 0.0098 | < 0.001 | 0.9258 | 0.2751 | P_Ce |
| AL_Pty | | | | | | 0.0168 | < 0.001 | 0.5059 | 0.2036 | AL_Pty |
| AL_Pt | | | | | | | 0.0440 | < 0.001 | 0.0034 | AL_Pt |
| P_Pt | | | | | | | | < 0.001 | < 0.001 | P_Pt |
| P_Yed | | | | | | | | | 0.1448 | P_Yed |







Figure 1



Figure 2

a


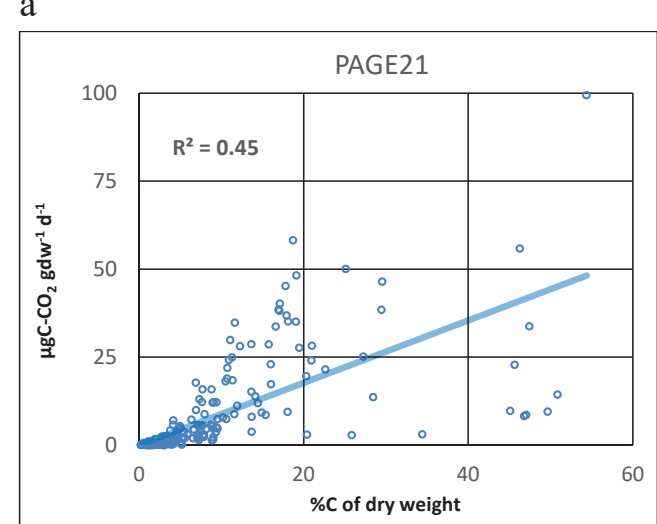

b

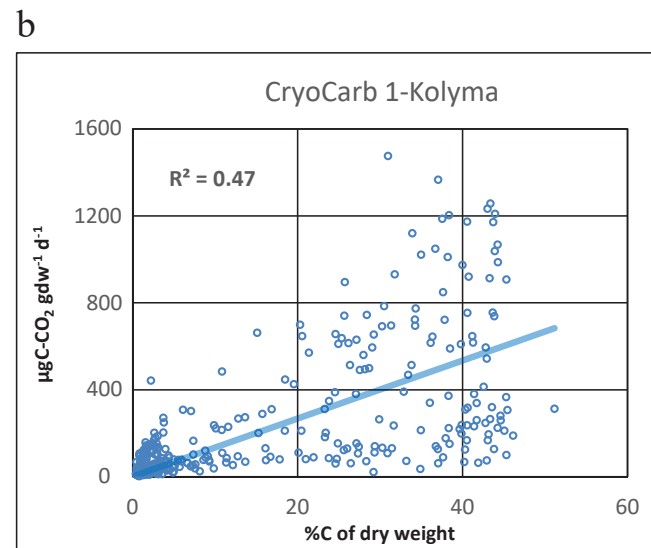






Figure 3

a

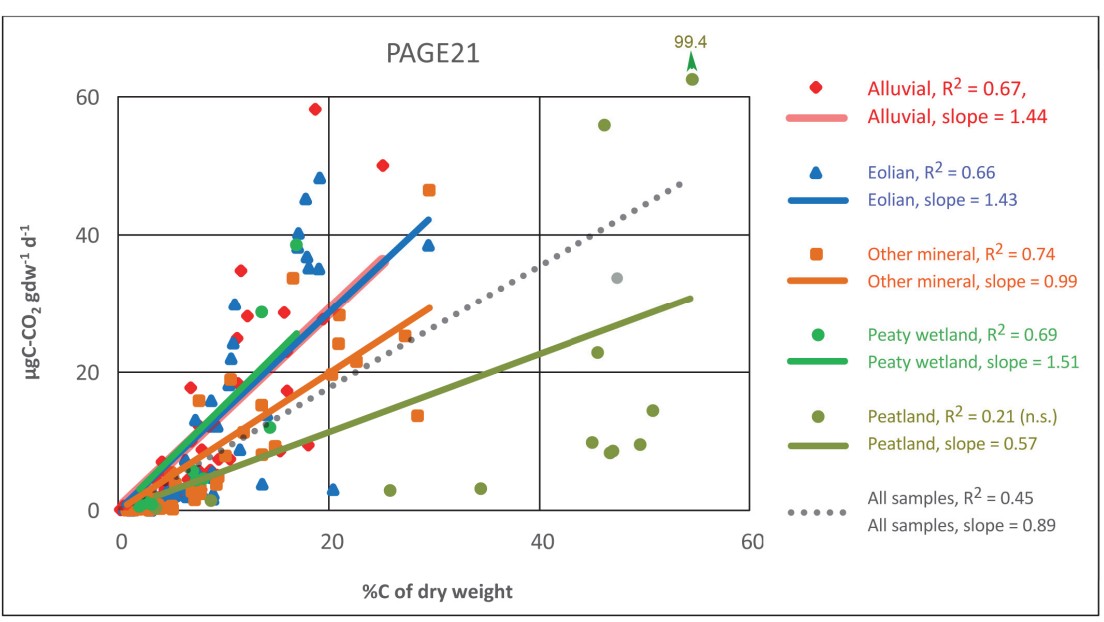

b

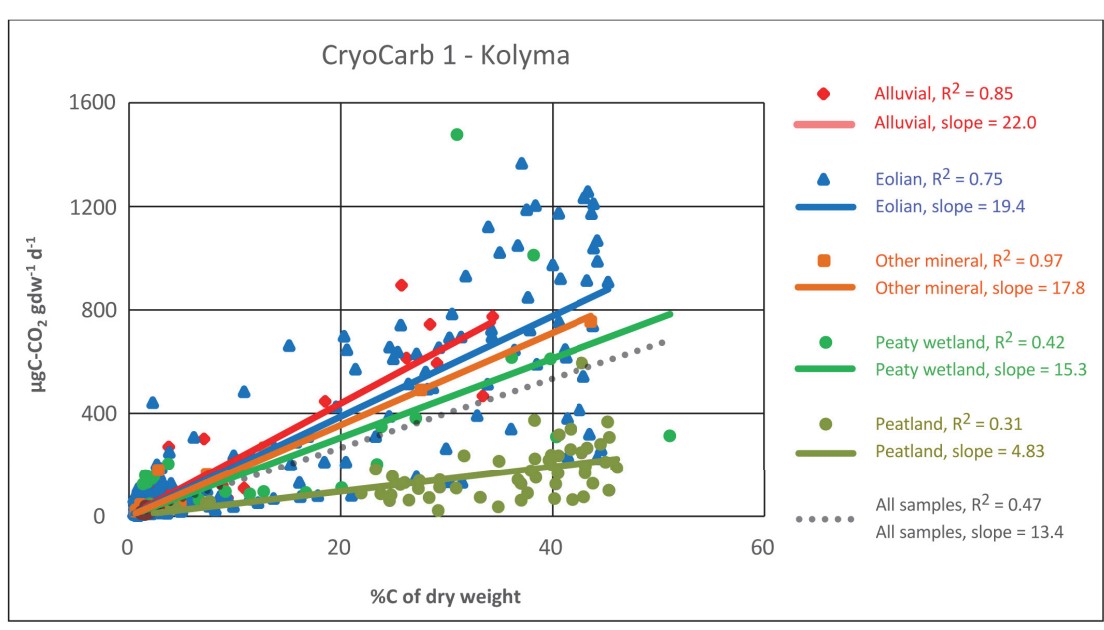




Figure 4





Figure 5.

a

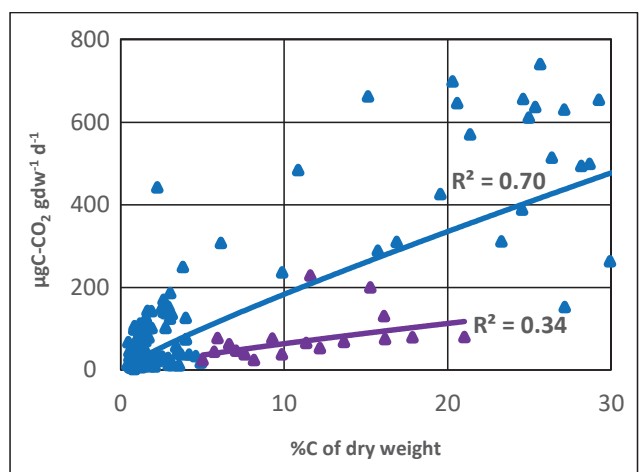


b

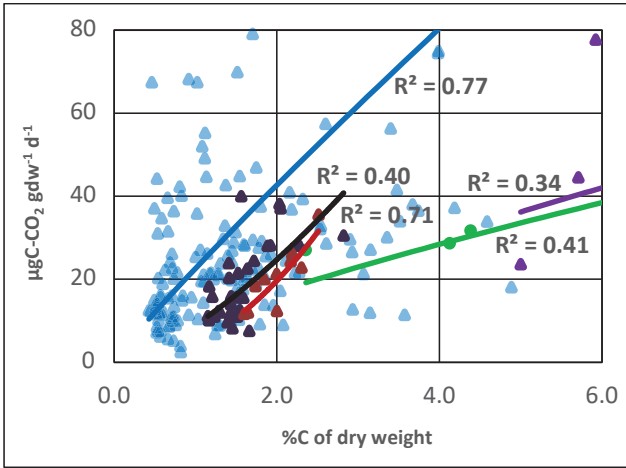



Figure 6.

a

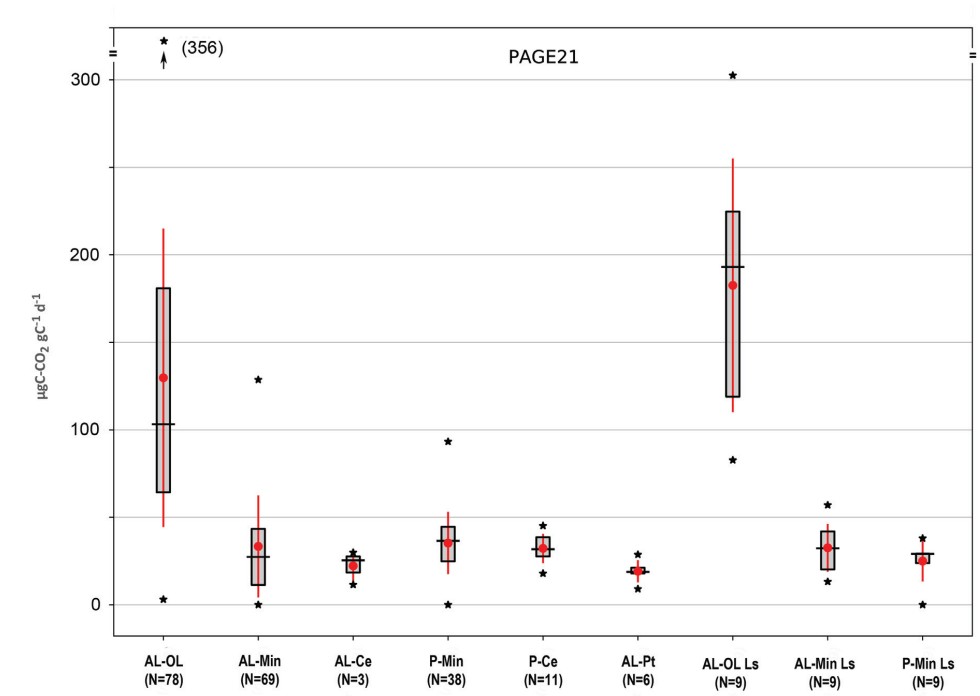


b

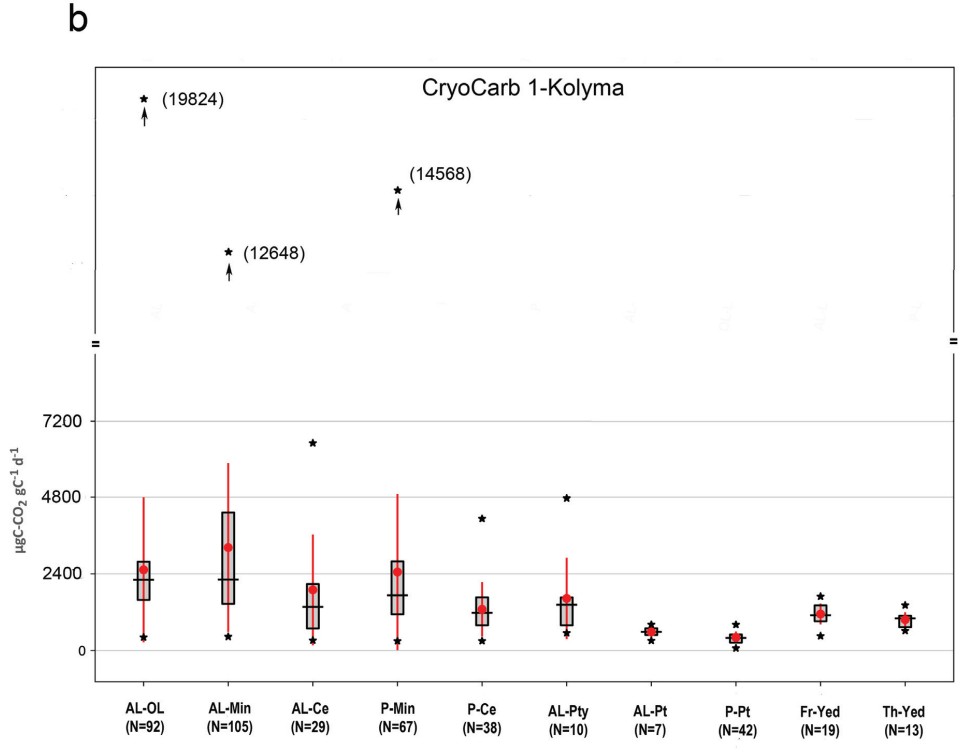