# Peer review of "Lability classification of soil organic matter in the northern permafrost region 1 2 3 Kuhry, P.1, Bárta, J.2, Blok, D.3, Elberling, B.4, Faucherre, S.4, Hugelius, G.1,5, Jørgensen, C. J.4,6, Richter, A.7, Šantrů"

_Biogeosciences, 2019_

## Referee Comment (RC1) · Anonymous Referee #1 · 7 Jun 2019

The manuscript 'Lability classification of soil organic matter in the northern permafrost region' by Kuhry and co-workers describes results from two large incubation experiments with soils from the circum-arctic region. The authors grouped their samples according to landscape units and further properties such as soil depth or carbon content and subsequently compared the $CO_2$ production between these groups. The manuscript is concerned with the very relevant question on organic carbon decomposition in northern, permafrost affected soils. It contributes to our understanding on how fast permafrost organic matter may be decomposed to the greenhouse gas $CO_2$. Furthermore, the authors present a novel concept by relating $CO_2$ production to different landscape units and soil classes that are represented in circum-arctic databases. The author's findings are novel and the conclusions drawn largely supported by the

presented data. The manuscript is well written and easy to read, however, there are some points that should be addressed before publication.

First of all, the authors compare data from two very different incubation experiments. While the PAGE21 experiment presents $CO_2$ production rates after almost one incubation year, the CryoCarb experiment(s) lasted only for four days. The difference in incubation time means that the different experiments give information about very different carbon pools. The authors are certainly aware that conceptual models divide soil organic matter into different pools characterized by different decomposition rate constants. This concept has also been applied to a wide variety of permafrost soils grouped into different classes, similar as in the current paper (Schädel et al., Circumpolar assessment of permafrost C quality and its vulnerability over time using long-term incubation data. Global Change Biology 2014; 20: 641-652). Latter paper demonstrates that all different kinds of soil (organic, surface mineral or deep mineral) the most rapidly degradable carbon pool represents only a very small fraction (< 5% of initial C). Hence, the CryoCarb experiment might give information about this small pool of rapidly cycling carbon in these soils but not on the overall 'available' pool. In contrast the PAGE21 experiment will likely give information on the much larger 'slow' carbon pool, which substantially differ in size between the different classes defined by Schädel et al.. The meaning of the data from the two incubation experiments for the different carbon pools should be clearly explained, e.g. in the introductin. The large dataset form the Schädel et al. paper in part supports the findings of the current manuscript (organic matter in deeper permafrost deposits is less degradable than surface organic matter) and in part contradict them (Schädel et al. find high decomposability of peat, the current paper not) and the results from this previous paper should be discussed as well.

Furthermore, the PAGE21 incubation follows a widely used approach, which makes the obtained data comparable to those from previous studies. In contrast, the CryoCarb experiment uses a novel approach, which appears to have some critical issues. First of all

the authors used samples dried at up to 50°C, which will most likely substantially harm if not kill the microbial community in these samples from very cold soils. The peak of CO2 production after rewetting and inoculation of these samples with viable organisms likely results from the rapidly degradable organic matter from (lysed) microorganisms. Also the study cited by the authors (Fierer and Schimel, 2003) concludes that this CO2 pulse does not come from the soil organic matter but from microbial carbon. In this case, what do the data of the very short CryoCarb incubation experiment mean for the decomposition of soil organic matter? Furthermore the authors should give some more information on the setup of the CryoCarb experiment and the preparation of the samples. If the samples were stored without freezing for up to two weeks before drying, a substantial fraction of the active carbon pool will have been decomposed before drying. Is it possible to account for this bias? And which inoculum was used for the different incubations? Were the samples incubated with inoculum from the same sampling site? And how was accounted for the carbon introduced by the inoculum? Was there a control incubation only with inoculum? Please explain this experiment in more detail and please also address the limitations of the CryoCarb experiment.

The different experiments present data from different sampling areas that were grouped into different classes. However, it remains unclear from which area samples are grouped into which class. It appears e.g. that all peat samples in the PAGE21 experiment originate from Storedalen Mire. To evaluate the significance of the data, it is important to know where the samples of the different classes come from and if they are representative for the whole landscape class they stand for. Therefore, I suggest presenting a table in the supplementary information with the origin of the samples grouped into the different landscape and soil classes.

The two experiments used a very different number of samples, and also the number of samples grouped in the different landscape and soil classes seem to be very different. I suggest presenting the statistical information (R2, p, n) including the number of data used for the presented regression analysis side by side, e.g. in the respective figures.

Furthermore, some of the datasets are fitted with a linear regression some with an exponential, power or polynomal regression but the reason for these different regression types are not explained. Some of the results are not reasonable even if the regression is statistically significant and the authors should give a comprehensible justification for the model they selected for fitting their data. Why should a linear increase of the C concentration result in an exponential increase of the $CO_2$ production?

One strength of the manuscript is that it relates $CO_2$ production to different landscape classes and soil materials. However, more and more classes and sub-classes are introduced for the two experiments and it gets more and more difficult to follow. I suggest that the authors critically review if all the different classes, sub-classes and groups of classes are required to come to the main conclusion of their manuscript.

One of the surprising results of this manuscript is that the decomposability of peat organic matter seems lower than that of organic matter in mineral soils, which partly contradicts previous findings. It is also surprising that the variability of $CO_2$ production from peat organic matter is very low (Fig. 6). I suggest giving this result more attention in the discussion. Do the samples represent peat from drained peat plateaus that are exposed to long-time of aerobic decomposition? How representative are the data for the Histosol/Histel class in the Northern Circumpolar Soil Carbon Database? What might be the reason for the low decomposability? Does this peat only represent ombrotrophic bogs or also minerotrophic fens?

The discussion is mainly considering previous studies of the authors working at the sites studied in the current manuscript. However, there is a wealth of recently published data from aerobic incubation studies considering a wide range of circum-arctic soils. To put the data of the current manuscript in a wider perspective I suggest stronger considering data from previous incubation studies from other sites, which might support or contradict the findings of the current manuscript.

The results section contains a substantial amount of discussion and also some description of methods, and I suggest to move this text to the respective section (see specific comments).

The line graphs (Fig. 2-Fig. 5) may be improved by using clearly different symbols and colors, and by inserting a legend to each of the panels.

specific comments:

L75: Should read 'permafrost zone' instead of 'thawing permafrost'. The estimate describe C fluxes from soils of the permafrost zone not of the thawing permafrost alone.

L 90: Schädel et al., 2014

L150ff: Please also specify how many replicates were used and how $CO_2$ production rates were measured.

L153: Please specify from which study areas the samples were collected.

L 157: Please specify from which sampling site the peat samples were collected.

L 168f: If only the rate after 363 days is considered in this manuscript it is not necessary to mention how often rates were measured (or cite the respective study of Faucherre, JGR, 2018; doi:10.1002/2017JG004069).

L197: I suggest not using 'available' in this context. The CryoCarb experiment give information on the very fast cycling C-pool, which is much smaller than the carbon pool that is available for microbial decomposition (see general comments).

L 205ff: This paragraph is mainly an explanation why the authors used %C as the main explanatory parameter for the measured $CO_2$ production rates. I suggest shortening this paragraph in the M&M section and shift the main part into the discussion, if necessary.

L285: Schädel et al., 2014

L282-291. A large part of this paragraph contains discussion and should be moved to

the respective section.

L 354f: Does this mean that CO2 production rates were similar in samples from peat deposits and mineral soils if C/N ratios were below 20? Please rephrase.

L357ff: please move to discussion.

L368 – 378: This paragraph rather describes methods and should go to the respective section, e.g. at the end of 2.5.

L378: Please explain to which subclasses you refer.

L420ff: Please explain what is meant by 'C-enriched' or 'organically-enriched'. To my understanding, every soil horizon that contains organic carbon is 'C-enriched'. How do the authors differentiate between 'C-enriched' and not 'C-enriched'?

L536ff: A large part of this paragraph belongs to the discussion, please move to the respective section.

Fig.2: I suggest omitting Fig. 2 since it gives the same data as Fig 3 including the fit of the total dataset.

Fig. 3: The different greens are difficult to differentiate. Please also use clearly different symbols.

Fig. 4 and Fig. 5: Please add legends to the figure and please use clearly different colors and symbols.

Fig. 6: Significant differences between the groups should be better indicated here and Table 4 could than go to the supplementary material

---

## Short Comment (SC1) · 7 Jun 2019

I find this manuscript a well written and valuable study addressing the vulnerability of permafrost carbon to climate change and the permafrost carbon feedback. It seems, however, that the authors assume recent permafrost warming as general knowledge. At the beginning of the introduction they could better provide proofs and estimates of this ground warming process and the relationship to the atmosphere. Recent publications on permafrost warming should enable also a direct comparison to some of the considered study sites.

---

## Referee Comment (RC2) · Anonymous Referee #2 · 1 Jul 2019

The authors report on the analysis of the organic carbon mineralization of soil and Yedoma material from numerous sites in the Northern hemisphere making use of two different incubation experiments. The authors cluster the samples into different source material (eolian, alluvial) and ecosystem / soil types (peatland, Turbels), aiming to provide estimates for the bioavailability of SOC in different Arctic terrestrial OC pools. As there is only scarce knowledge on the vulnerability of the tremendous OC pools in the Arctic, the overall objective of the manuscript to come up with such estimates is of great interest especially to refine carbon modelling. Using a large data set of OC mineralization rates/data is a very straightforward approach to obtain estimates for the potential bioavailability of OC. However, the manuscript appears a bit like the attempt of a group of authors to get a manuscript out of existing data sets using correlations

of the least common multiples which are stated to be %C, C/N and bulk density. The explanation why a single day mineralization rate at the end of a long-term incubation, and a short-term incubation go together is questionable. After the rewetting of dried material it is known that the first flush of $CO_2$ within the first days is mainly derived from OC additionally available due to the physical impacts of the drying (disintegration of SOM, lysed microorganisms) especially as it was done at higher temperature. Given the highly seasonal DOC content in permafrost affected soils (the material presumably mainly driving the $CO_2$ evolution), this short term incubation is also more like a snapshot in time. The authors should explain much better why they use these two incubations, and what oven dried inoculated vs. fresh material can tell us about the bioavailability of soil organic matter under natural conditions. Furthermore, it would be interesting if the authors give the cumulative OC over the full period of the long term incubation. Besides these technical aspects, the manuscript appears very descriptive. There is a number of studies on the distribution and composition of OC in permafrost affected soils that demonstrate possible OM vulnerability to increased microbial decay. It would be interesting to discuss the data in more detail especially in view of the composition of the OM, even it would just be C/N ratios as given by the authors. line 195-201 - The drying-rewetting of this approach lead to an increased respiration due to lysed cells, physical breakdown of soil material etc. Thus it may serve as a proxy for potential amount of 'artificially' labile OC, but does not reflect the natural amount of labile OC. Detailed comments: line 309-310 - Something to be expected, the more substrate the higher the respiration. But it neglects all other factors driving C-release, like pH etc. line 317-319 - If I got your M&M section right, you measured the long term incubation samples at one point in time after almost a year. Of course its much lower, the short term got a higher $CO_2$ due to rewetting effects plus the flush in mostly labile OM, and the long run incubation represents more stable OM mioties. How is the cummulative OC release in the long term experiment, and thus the overall OC release? line 572-573 - This is normal, you have in most soils systems not matter if arctic, temperate or tropic an exponential decay of the respiration rates. For the long

term incubation the total amount of released C would be interesting. line 575-577 - Didn't you state before that it is not possible to compare the mineralization rates due to the different sampling, sample treatment and incubation? line 578-579 - How did you come up with this assumption? What makes the data robust? line 581-582 - You may be able to relate the studied soil samples to larger scale OC inventories, but how do the lab incubations relate to the natural systems with differing pH, active layer depth, soil humidity etc.? line 619-629 - How does this deep OC rather stable OM relate to C/N ratios? line 632 - Please use a other word than "restistance", SOM does not "actively" resist decomposition/mineralization. line 632-633 - There is already some work trying to elucidate the underlying mechanisms on SOM stabilization in permafrost affected soils (e.g. Gentsch et al. EJSS 2015; Mueller et al. GCB 2015). line 633-643 - Besides a solely microbial driven decomposition, there are also some more soil physical and chemical constraints to SOM mineralization (see comment above). line 637-643 - Peat decomposition is dominated by the water regime. Drained peatlands can loose substantial amounts of OC on very short timescales. Thus, this only explains retarted decomposition in intact peatlands, not so much in other peat-like soil materials. line 644-657 - In natural systems such short term flushes are known to happen very often (freeze-thaw; drying-rewetting), thus for the labile OC the short term incubations gives for one moment in time (sampling date) a good insight. For a more solid OM material proxy the long term incubation is still of some use, but it would be nice to get either the overall OC and not just a rate at day x, or k-values for the long term decay curves. line 661-663 - This holds true for most soils, amount of substrate means low DBD, this linked to N availability determines OC mineralization. I would have wondered if its different in colder soils. line 664-667 - What about other proxies like pH? line 668-669 - Do you have other soil parameters that could be used to fine tune the multiple regressions?

---

## Author Comment (AC1) · 4 Sep 2019

**BG-2019-89, Authors response**

**Reviewer #1**

**General comments:**

The manuscript 'Lability classification of soil organic matter in the northern permafrost region' by Kuhry and co-workers describes results from two large incubation experiments with soils from the circum-arctic region. The authors grouped their samples according to landscape units and further properties such as soil depth or carbon content and subsequently compared the $CO_2$ production between these groups. The manuscript is concerned with the very relevant question on organic carbon decomposition in northern, permafrost affected soils. It contributes to our understanding on how fast permafrost organic matter may be decomposed to the greenhouse gas $CO_2$. Furthermore, the authors present a novel concept by relating $CO_2$ production to different landscape units and soil classes that are represented in circum-arctic databases. The author's findings are novel and the conclusions drawn largely supported by the presented data. The manuscript is well written and easy to read, however, there are some points that should be addressed before publication.

*AR1: We would like to thank the reviewer for the general positive appraisal of our study and constructive comments*

First of all, the authors compare data from two very different incubation experiments. While the PAGE21 experiment presents $CO_2$ production rates after almost one incubation year, the CryoCarb experiment(s) lasted only for four days. The difference in incubation time means that the different experiments give information about very different carbon pools. The authors are certainly aware that conceptual models divide soil organic matter into different pools characterized by different decomposition rate constants. This concept has also been applied to a wide variety of permafrost soils grouped into different classes, similar as in the current paper (Schädel et al., Circumpolar assessment of permafrost C quality and its vulnerability over time using long-term incubation data. Global Change Biology 2014; 20: 641-652). Latter paper demonstrates that all different kinds of soil (organic, surface mineral or deep mineral) the most rapidly degradable carbon pool represents only a very small fraction (< 5% of initial C). Hence, the CryoCarb experiment might give information about this small pool of rapidly cycling carbon in these soils but not on the overall 'available' pool. In contrast the PAGE21 experiment will likely give information on the much larger 'slow' carbon pool, which substantially differ in size between the different classes defined by Schädel et al. The meaning of the data from the two incubation experiments for the different carbon pools should be clearly explained, e.g. in the introduction. The large dataset from the Schädel et al. paper in part supports the findings of the current manuscript (organic matter in deeper permafrost deposits is less degradable than surface organic matter) and in part contradict them (Schädel et al. find high decomposability of peat, the current paper not) and the results from this previous paper should be discussed as well.

*AR2: We are aware that previous incubation studies have subdivided SOM into conceptual pools, representing groups of organic compounds with different chemical recalcitrance and decomposition rate constants. We consider such an approach not relevant to our study because we are dealing with two very different laboratory incubation experiments with different sample pre-treatment and measuring $CO_2$ release at only two distinct time periods under different conditions of temperature and moisture. We present absolute flux rates on lines 561-571 of the submitted paper, but caution against a direct comparison of these values. Instead we address the relative lability of the different*

*groups of samples recognized in these incubation experiments, which should be less sensitive to various sample pre-treatment and incubation setups*

*Nonetheless, we appreciate the comments by the reviewer and we will emphasize in the introduction that the CryoCarb experiments most likely address the 'fast' (Schädel et al., 2014) and 'labile' (Knoblauch et al., 2013) SOM pools, which represent a small fraction of the total pool and decompose within a (few) year(s), whereas the PAGE21 experiment mostly addresses the 'slow' and 'stable' SOM pools, with C cycling typically within a (few) decade(s), in these respective studies*

*Additional reference:*

*Knoblauch, C. et al., 2013. Predicting long-term carbon mineralization and trace gas production from thawing permafrost of Northeast Siberia. Global Change Biology, 19: 1160–1172*

*(For peat decay see further down)*

Furthermore, the PAGE21 incubation follows a widely used approach, which makes the obtained data comparable to those from previous studies. In contrast, the CryoCarb experiment uses a novel approach, which appears to have some critical issues. First of all, the authors used samples dried at up to 50 °C, which will most likely substantially harm if not kill the microbial community in these samples from very cold soils. The peak of $CO_2$ production after rewetting and inoculation of these samples with viable organisms likely results from the rapidly degradable organic matter from (lysed) microorganisms. Also the study cited by the authors (Fierer and Schimel, 2003) concludes that this $CO_2$ pulse does not come from the soil organic matter but from microbial carbon. In this case, what do the data of the very short CryoCarb incubation experiment mean for the decomposition of soil organic matter?

*AR3: On line 648 of the submitted manuscript we acknowledge that microbial necromass is likely an important component of the $CO_2$ flux observed in the short-term CryoCarb incubation experiments. Certainly, a drying at ≤ 50 °C and subsequent rewetting of the sample will kill (part of) the microbial community*

*We do not expect that this would significantly bias our approach, which is based on the so-called Birch effect (Birch, 1958), showing that after a dry/wet cycle $CO_2$ mineralization increases. The more severe the shock, the larger amount of C is released. The extra C originates from mineralization of available C released from organo - mineral complexes and died biomass. In our sample pretreatment with rapid drying at ≤ 50 °C we expect that a larger part of biomass died and decomposed already during this process, which should therefore not severely affect our later measurements. Fierer and Schimel (2003) showed that a substantial part of the released C can also come from microbial biomass which died due to the osmotic shock after rewetting of soil. However, in their case, samples were dried at room temperature resulting in less of a shock in the drying process to the microbial community. Furthermore, the original microbial biomass is expected to be proportional to the amount of degradable SOM (e.g. Capek et al., 2015), so that microbial necromass also provides an indirect measurement of the size of the fast degradable pool. Our measurements can be affected by limitation of C mineralization due to the small size of surviving biomass, which we overcome by inoculation with living cells. The principle of the Birch Effect is still used in ecological studies ranging from large scale carbon cycling in ecosystems to detailed studies of SOC availability (e.g. Jarvis et al., 2017). As written in methods (lines 195-198): we consider that "It is well documented that extra C is released after rewetting of dry soil, the amount of which is site and soil type specific and represents available fraction of soil C (e.g. Franzluebbers et al., 2000; Santruckova et al., 2006)".*

*Additional references:*

*Birch, H. F., 1958. The effect of soil drying on humus decomposition and nitrogen availability. Plant and Soil, 10.1: 9-31*

*Capek, P. et al., 2015. The effect of warming on the vulnerability of subducted organic carbon in arctic soils. Soil Biology and Biochemistry, 90: 19-29*

*Jarvis, P. et al., 2017. Drying and wetting of Mediterranean soils stimulates decomposition and carbon dioxide emission: the "Birch effect". Tree Physiology, 27: 929–940*

Furthermore, the authors should give some more information on the setup of the CryoCarb experiment and the preparation of the samples. If the samples were stored without freezing for up to two weeks before drying, a substantial fraction of the active carbon pool will have been decomposed before drying. Is it possible to account for this bias? And which inoculum was used for the different incubations? Were the samples incubated with inoculum from the same sampling site? And how was accounted for the carbon introduced by the inoculum? Was there a control incubation only with inoculum? Please explain this experiment in more detail and please also address the limitations of the CryoCarb experiment.

*AR4: The CryoCarb-Kolyma and CryoCarb-Taymyr samples were stored in a ground pit dug into the active layer for up to two weeks, before further processing. We consider that active layer samples would have been little impacted by this storage under 'natural' conditions, but acknowledge that (some of) the gradually thawing permafrost layer samples might have experienced some initial decay. Due to the varying duration of storage it is difficult to account for any bias, except to accept that this introduces a certain noise in our assessments*

*We will add this and the following clarifications on the CryoCarb experiments to the text:*

*We failed to specify explicitly in the submitted paper that the pre-treatment of samples for the CryoCarb-Seida experiment differed from the other two CryoCarb experiments because collected samples stored under field conditions were subsequently kept in frozen storage for c. 10 years (see Table S1), before further processing. We will insert this statement in line 186 of the submitted paper*

*For each incubated sample, 0.2g of dry soil was inoculated with 0.003-0.008g of dry soil inoculum in 1.6 ml of water (soil:$H_2O$, 1:100, weight/volume). We consider that the small dry weight of our soil inoculi (which, in turn, have ≤2% microbial biomass) has no significant impact on our C release measurements. The viability of inoculi was checked by incubation in water and measuring its respiration. We will insert these statements in line 186 of the submitted paper*

*As for the inoculi, we used layer specific composite inoculi (see lines 190-192) collected in each study area for each CryoCarb experiment separately. We will further clarify this statement in the revision*

*We have emphasized in our submitted paper (lines 199 and 555), that different sample pre-treatment, drying/wetting, and the use of different inoculi in the CryoCarb experiments makes it difficult to compare CryoCarb experiments among each other, and certainly with the longer-term PAGE21 experiment. We stress, therefore, the relative lability of samples instead of absolute $CO_2$ release rates, and consider that the CryoCarb experiments provide useful information on differences in early decay rates corresponding largely to the fast/labile SOM pool in the samples. Furthermore, we acknowledge that in longer term experiment such as the PAGE21 incubation, carried out under strict protocols, conditions in the incubated material become increasingly more artificial over time potentially affecting measured decay rates (lines 651-655)*

The different experiments present data from different sampling areas that were grouped into different classes. However, it remains unclear from which area samples are grouped into which class. It appears e.g. that all peat samples in the PAGE21 experiment originate from Stordalen Mire. To evaluate the significance of the data, it is important to know where the samples of the different classes come from and if they are representative for the whole landscape class they stand for. Therefore, I suggest presenting a table in the supplementary information with the origin of the samples grouped into the different landscape and soil classes.

*AR5: We explain in lines 247-249, which of the recognized landscape classes are represented in each incubation experiment. As requested, we will add a Table to Supplementary Materials, summarizing once more which classes are represented in which incubation experiment/study area*

The two experiments used a very different number of samples, and also the number of samples grouped in the different landscape and soil classes seem to be very different. I suggest presenting the statistical information ($R_2$, p, n) including the number of data used for the presented regression analysis side by side, e.g. in the respective figures.

*AR6: $R_2$ values are already indicated in all figures. Furthermore, the p value (we use a simple cutoff of 0.05) is referred to in the figure caption, emphasizing the few cases where the regression is not significant. We propose to add n=number of cases to the figures in the revision of the paper*

Furthermore, some of the datasets are fitted with a linear regression some with an exponential, power or polynomial regression but the reason for these different regression types are not explained. Some of the results are not reasonable even if the regression is statistically significant and the authors should give a comprehensible justification for the model they selected for fitting their data. Why should a linear increase of the C concentration result in an exponential increase of the $CO_2$ production?

*AR7: On lines 321-322 we indicate that for the sake of simplicity we apply linear regressions to the datasets depicted in figures 2-3 and S3-S5. These are very large datasets subdivided into many groups and figures would become too complex to easily understand if a variety of regression fits would be applied. However, when considering further subdivisions separately (figure 4), we applied polynomial or exponential functions that provided better fits (lines 369, 377 and 389). These regressions show that samples with very high %C show relatively high decay rates, which can easily be explained by the fact that they most often correspond to top soil organic samples in mineral soils or peat surface samples in peatlands that still contain a relatively fast SOM pool in the recently deposited litter (lines 383, 389-390 and 396)*

One strength of the manuscript is that it relates $CO_2$ production to different landscape classes and soil materials. However, more and more classes and sub-classes are introduced for the two experiments and it gets more and more difficult to follow. I suggest that the authors critically review if all the different classes, sub-classes and groups of classes are required to come to the main conclusion of their manuscript.

*AR8: We consider that a further subdivision of the samples is useful (section 3.3 and figure 4), because on the one hand it permits a more detailed analysis of decay rates by introducing best fits (see above), and on the other hand it shows that certain patterns are consistent across landscape units. For instance, samples that experienced recent minerogenic inputs in eolian, alluvial, colluvial and wetland settings display high relative lability, whereas deeper C-enriched cryoturbated samples in mineral soils show consistently lower rates and no rapid increases at higher %C. This suggests important soil processes that affect decay rates and need further study (see Discussion). While subdivisions are useful, these classes are currently not represented in any global circumpolar database and therefore of limited use for upscaling*

One of the surprising results of this manuscript is that the decomposability of peat organic matter seems lower than that of organic matter in mineral soils, which partly contradicts previous findings. It is also surprising that the variability of $CO_2$ production from peat organic matter is very low (Fig. 6). I suggest giving this result more attention in the discussion. Do the samples represent peat from drained peat plateaus that are exposed to long-time of aerobic decomposition? How representative are the data for the Histosol/Histel class in the Northern Circumpolar Soil Carbon Database? What might be the reason for the low decomposability? Does this peat only represent ombrotrophic bogs or also minerotrophic fens?

*AR9: The Stordalen Mire samples in the PAGE21 experiment are from a palsa complex, with profiles from both palsas and fens without permafrost. While we have not analyzed the peat composition in the palsa profile, the deposit in these sites is normally formed by fen peat with a thin top layer of dry palsa peat and permafrost that formed epigenetically (most often during the Little Ice Age). This site can be considered quite typical for permafrost peatlands in the isolated/sporadic permafrost zones. The Seida peat profiles are from peat plateaus and interspersed permafrost-free fens in the discontinuous permafrost zone. The Taymyr and Shalaurovo sites are from polygonal wetlands, whereas the Cherskij sites include permafrost peatlands developed in Alasses (drained thermokarst basins). All together, we consider they represent an adequate sample of the Histels/Histosols in the permafrost region. Table S1 includes the permafrost zone of the different study areas and we consider this enough information to highlight that our peat samples come from a variety of permafrost settings*

*AR10: With regard to peat decay, we do not consider that our results necessarily contradict the findings of Schädel et al. (2014). This study recognized a group of samples from organic soils (> 20% initial C), ranging in depth between 0 and 120 cm. We consider that this group will include both top soil organic samples in mineral soils, as well as deeper peat deposits. At the same time, according to Schädel et al. (2014), these organic soil samples 'showed the largest range in slow C pool sizes', and 'some soils being especially vulnerable, showing high potential C losses ...', whereas deeper organic soil samples 'were less likely to respire large amounts of within the 50 year time frame'. While we cannot be sure without a proper division of the organic soils in Schädel et al., we suggest that both studies might show the same trends. As stated by the reviewer, our deep mineral samples from Yedoma show the same 'relatively low' lability as in Schädel et al.*

*In our revisions, we will compare our results to those in Schädel et al. and some other studies (eg. Gentsch et al., 2015b; incubation of Turbel soils). Unfortunately, there are few studies that compare decay rates across such a wide spectrum of soil/deposit types for the permafrost region as presented in this study*

*Additional reference:*

*Gentsch, N. et al., 2015b. Properties and bioavailability of particulate and mineral-associated organic matter in Arctic permafrost soils, Lower Kolyma Region, Russia. European Journal of Soil Science, 66: 722-734*

The discussion is mainly considering previous studies of the authors working at the sites studied in the current manuscript. However, there is a wealth of recently published data from aerobic incubation studies considering a wide range of circum-arctic soils. To put the data of the current manuscript in a wider perspective I suggest stronger considering data from previous incubation studies from other sites, which might support or contradict the findings of the current manuscript.

*See above*

The results section contains a substantial amount of discussion and also some description of methods, and I suggest to move this text to the respective section (see specific comments).

*See below*

The line graphs (Fig. 2-Fig. 5) may be improved by using clearly different symbols and colors, and by inserting a legend to each of the panels.

*AR11: Figures 3-5 and S3-S5 are very data dense and include multiple classes. Panels are already inserted in figures 3 and S3-5. We will add panels to figures 4 and 5, and aim to further optimize the use of colors and symbols*

**Specific comments:**

L75: Should read 'permafrost zone' instead of 'thawing permafrost'. The estimate describe C fluxes from soils of the permafrost zone not of the thawing permafrost alone.

*Yes, thank you*

L 90: Schädel et al., 2014

*OK*

L150ff: Please also specify how many replicates were used and how $CO_2$ production rates were measured.

*This study did not include replicate samples. The measurement technique is shortly described on line 170. Full details are available from Faucherre et al. (2018)*

L153: Please specify from which study areas the samples were collected.

*This information will be conveyed in an additional Table S2 (see above)*

L 157: Please specify from which sampling site the peat samples were collected.

*Stordalen Mire will be added to the text*

L 168f: If only the rate after 363 days is considered in this manuscript it is not necessary to mention how often rates were measured (or cite the respective study of Faucherre, JGR, 2018; doi:10.1002/2017JG004069).

*We will cite again the Faucherre et al. (2018) paper*

L197: I suggest not using 'available' in this context. The CryoCarb experiment give information on the very fast cycling C-pool, which is much smaller than the carbon pool that is available for microbial decomposition (see general comments).

*The text on line 197 is a very general statement, but we will change to 'an indicator for the very fast cycling C pool (instead of SOM lability)', on line 196*

L 205ff: This paragraph is mainly an explanation why the authors used %C as the main explanatory parameter for the measured $CO_2$ production rates. I suggest shortening this paragraph in the M&M section and shift the main part into the discussion, if necessary.

*We consider the location of this text appropriate, since it pertains a discussion concerning the best methodological approach. The main discussion in this paper addresses the observed relative lability differences of the landscape classes, not methodological issues*

L285: Schädel et al., 2014

*OK*

L282-291. A large part of this paragraph contains discussion and should be moved to the respective section.

*This paragraph includes the reason for and results of regressions of C release against selected geochemical parameters, and then obvious patterns are discussed. Again, the main discussion in this paper addresses the observed relative lability differences of the landscape classes, not methodological issues*

L 354f: Does this mean that $CO_2$ production rates were similar in samples from peat deposits and mineral soils if C/N ratios were below 20? Please rephrase.

*Yes, this is evident from how the peat regression intercepts the other regressions at C/N values ≤20 (see Fig. S4). We will clarify this statement in the revision*

L357ff: please move to discussion.

*We consider this a clear result, supported by previous studies. No need to discuss it later on*

L368 – 378: This paragraph rather describes methods and should go to the respective section, e.g. at the end of 2.5.

*Yes, we agree*

L378: Please explain to which subclasses you refer.

*It pertains to all subclasses recognized in Figures 4a-c. This will be clarified in the revision*

L420ff: Please explain what is meant by 'C-enriched' or 'organically-enriched'. To my understanding, every soil horizon that contains organic carbon is 'C-enriched'. How do the authors differentiate between 'C-enriched' and not 'C-enriched'?

*This is explained in lines 425-427. We only consider a deeper soil sample carbon(organically)-enriched if it has at least two times the %C of the directly adjacent mineral subsoil samples. The actual values can actually vary from study area to study area, or even from profile to profile, and depends among others on soil texture (Palmtag and Kuhry, 2018)*

*Additional reference:*

*Palmtag, J. and Kuhry, P. 2018. Grain size controls on cryoturbation and soil organic carbon density in permafrost-affected soils. Permafrost and Periglacial Processes, 29: 112–120*

L536ff: A large part of this paragraph belongs to the discussion, please move to the respective section.

*We consider these results, supported by previous studies. No need to discuss it later on*

Fig.2: I suggest omitting Fig. 2 since it gives the same data as Fig 3 including the fit of the total dataset.

*OK, Figure 2 can be considered redundant and will be removed*

Fig. 3: The different greens are difficult to differentiate. Please also use clearly different symbols.

*We will aim to improve colors and symbols*

Fig. 4 and Fig. 5: Please add legends to the figure and please use clearly different colors and symbols.

*Yes*

Fig. 6: Significant differences between the groups should be better indicated here and Table 4 could than go to the supplementary material

*We considered that, but there are too many sample groups and too many combinations of significant differences that would make the figure difficult to grasp. We, therefore, opted for a separate table*

**Reviewer #2**

**General comments:**

The authors report on the analysis of the organic carbon mineralization of soil and Yedoma material from numerous sites in the Northern hemisphere making use of two different incubation experiments. The authors cluster the samples into different source material (eolian, alluvial) and ecosystem / soil types (peatland, Turbels), aiming to provide estimates for the bioavailability of SOC in different Arctic terrestrial OC pools. As there is only scarce knowledge on the vulnerability of the

tremendous OC pools in the Arctic, the overall objective of the manuscript to come up with such estimates is of great interest, especially to refine carbon modelling. Using a large data set of OC mineralization rates/data is a very straightforward approach to obtain estimates for the potential bioavailability of OC. However, the manuscript appears a bit like the attempt of a group of authors to get a manuscript out of existing data sets using correlations of the least common multiples which are stated to be %C, C/N and bulk density.

*AR11: There have been no previous studies comparing (relative) SOM lability across such a large diversity of soil/deposit types, representing all major SOC pools in the northern permafrost region. In defining these landscape classes, we have used those represented in large northern circumpolar databases like the NCSCD, making our results available for upscaling. The CryoCarb datasets were previously unpublished, whereas the grouping for the PAGE21 dataset is different from that in Faucherre et al. (2018)*

The explanation why a single day mineralization rate at the end of a long-term incubation, and a short-term incubation go together is questionable. After the rewetting of dried material it is known that the first flush of $CO_2$ within the first days is mainly derived from OC additionally available due to the physical impacts of the drying (disintegration of SOM, lysed microorganisms) especially as it was done at higher temperature. Given the highly seasonal DOC content in permafrost affected soils (the material presumably mainly driving the $CO_2$ evolution), this short term incubation is also more like a snapshot in time. The authors should explain much better why they use these two incubations, and what oven dried inoculated vs. fresh material can tell us about the bioavailability of soil organic matter under natural conditions.

*AR12: We want to compare if the relative SOM lability in different landscape classes is consistent across time of incubation, by comparing the short term CryoCarb experiments to the longer term PAGE21 experiment. As stated in AR2 to Reviewer #1, we will clarify that the CryoCarb experiments mostly concerns the 'fast'/'labile' SOM pool, whereas the PAGE21 experiment represents the 'slow'/'stable' pool (Knoblauch et al., 2013; Schädel et al., 2014)*

*Furthermore, we acknowledge on line 648 of the submitted manuscript that microbial necromass is likely an important component of the $CO_2$ flux observed in the short-term CryoCarb incubation experiments. We comment extensively on this issue in AR3 to Reviewer #1, who also addressed this aspect of the CryoCarb experiment*

Furthermore, it would be interesting if the authors give the cumulative OC over the full period of the long term incubation.

*AR13: We have decided to compare two snap-shots in time to assess if the relative SOM lability in soils/deposits of different landscape classes remains consistent over time. Cumulative releases over one year in the PAGE21 experiment are addressed in Faucherre et al. (2018). On lines 592-594 in the submitted paper we make reference to the low cumulative release over one year from peat deposits in the PAGE21 experiment to corroborate the low relative lability of peat consistently observed in our different incubation experiments*

Besides these technical aspects, the manuscript appears very descriptive. There is a number of studies on the distribution and composition of OC in permafrost affected soils that demonstrate possible OM vulnerability to increased microbial decay. It would be interesting to discuss the data in more detail especially in view of the composition of the OM, even it would just be C/N ratios as given by the authors.

*AR14: We discuss the value of C/N as a proxy for SOM decomposability in the methods section (lines 205-232). We then confirm in results that using C/N as a geochemical parameter gives very similar results to %C (e.g. Fig. S4). Furthermore, we highlight some peculiarities of the C/N relationships in the results and discussion section (lines 355, 357 and 642). However, the main purpose of the paper is to assess whether there are consistent differences in relative SOM lability over incubation conditions*

*and time referring to the major SOC pools in the northern permafrost region. This shows that Histels/Histosols, C-enriched cryoturbated material in Turbels and Yedoma Ice Complex deposits, which together represent ≥50% of the total permafrost region C pool, display low lability (lines 686-692). We consider this an important result*

*Nonetheless, we can agree with the reviewer that the underlying soil (microbial) processes underlying these observations urgently require more research. This pertains to the role of organo-mineral associations in protecting SOM from decomposition as well as the low mineralization rates in peat. With regard to the former, we will extend our discussion of Turbels with adding results from studies by Gentsch et al. (2015b, 2018), that showed that most of OM in Turbels is associated with mineral material and displays low lability in incubation experiments (which is corroborated by our results)*

*Extra reference:*

*Gentsch, N. et al., 2018. Temperature response of permafrost soil carbon is attenuated by mineral protection. Global Change Biology, 24: 3401–3415*

Line 195-201 - The drying-rewetting of this approach lead to an increased respiration due to lysed cells, physical breakdown of soil material etc. Thus it may serve as a proxy for potential amount of 'artificially' labile OC, but does not reflect the natural amount of labile OC.

*See AR3 and AR12*

**Detailed comments:**

Line 309-310 - Something to be expected, the more substrate the higher the respiration. But it neglects all other factors driving C-release, like pH etc.

*Our datasets did not have consistent information on pH, Fe, clay fraction, soil moisture, etc and these could therefore not be considered in our analyses. Much variability remains in our regressions, even after partitioning into our landscape classes, which has to be explained by other local soil factors. We will add statements in this regard to the discussion and conclusions*

Line 317-319 - If I got your M&M section right, you measured the long term incubation samples at one point in time after almost a year. Of course its much lower, the short term got a higher $CO_2$ due to rewetting effects plus the flush in mostly labile OM, and the long run incubation represents more stable OM mioties. How is the cumulative OC release in the long term experiment, and thus the overall OC release?

*In our submitted paper we refer to the PAGE21 measurement after nearly one year, for reasons explained above (AR2 and AR12). Furthermore, we refer to the (low) cumulative release in peat compared to mineral soil samples observed in this experiment (AR13). Faucherre et al. (2018) provides a full account of the PAGE21 experiment, including measurements at 5 time periods as well as cumulative releases*

line 572-573 - This is normal, you have in most soils systems not matter if arctic, temperate or tropic an exponential decay of the respiration rates. For the long term incubation the total amount of released C would be interesting.

*See above*

line 575-577 – Didn't you state before that it is not possible to compare the mineralization rates due to the different sampling, sample treatment and incubation?

*Yes, we stress here that direct comparisons of the magnitude of the fluxes are difficult. We will clarify that the sample pre-treatment of the Seida samples (10 yr in freezer before further processing) was different than in the Kolyma and Taymyr samples (drying directly after the field period)*

line 578-579 - How did you come up with this assumption? What makes the data robust?

*We have shown that the relative SOM lability of landscape classes is consistent among experiments (and for different geochemical parameters and C release units). We, therefore, conclude that the relative SOM lability ranking is robust*

line 581-582 - You may be able to relate the studied soil samples to larger scale OC inventories, but how do the lab incubations relate to the natural systems with differing pH, active layer depth, soil humidity etc.?

*As explained in our answer for your comment on lines 309-310, we agree that local factors such as pH, soil moisture, etc are important but could not be analyzed with our datasets. Active layer aspects are shown in figure 6*

line 619-629 - How does this deep OC rather stable OM relate to C/N ratios? line 632

*The C/N ratio of peat decreases with depth, as a result of preferential loss of C during the decomposition process (Kuhry and Vitt, 1996). Results in figure S4 show that when the C/N of peat becomes ≤20, C release rates become similar to those observed in mineral subsoil samples with similar C/N ratios*

- Please use another word than "resistance", SOM does not "actively" resist decomposition/ mineralization.

*We will change this statement to '…, can be expected to show low rates of decomposition'*

line 632-633 - There is already some work trying to elucidate the underlying mechanisms on SOM stabilization in permafrost affected soils (e.g. Gentsch et al. EJSS 2015; Mueller et al. GCB 2015).

*We will extend the discussion on the role of mineral-organic associations in Turbels with results presented in Gentsch et al., 2015b, 2018*

line 633-643 – Besides a solely microbial driven decomposition, there are also some more soil physical and chemical constraints to SOM mineralization (see comment above).

*See above*

line 637-643 - Peat decomposition is dominated by the water regime. Drained peatlands can loose substantial amounts of OC on very short timescales. Thus, this only explains retarded decomposition in intact peatlands, not so much in other peat-like soil materials.

*Yes, peatlands can become wetter or drier as a result of permafrost degradation. Dried out peat can decompose quickly, but is also susceptible to (deep) peat fires resulting in very rapid C releases to the atmosphere. We will add a statement after line 643*

Line 644-657 - In natural systems such short term flushes are known to happen very often (freeze-thaw; drying-rewetting), thus for the labile OC the short term incubations gives for one moment in time (sampling date) a good insight. For a more solid OM material proxy the long term incubation is still of some use, but it would be nice to get either the overall OC and not just a rate at day x, or k-values for the long term decay curves.

*C release rates over 5 measurement periods in one year are presented in Faucherre et al., 2018*

line 661-663 - This holds true for most soils, amount of substrate means low DBD, this linked to N availability determines OC mineralization. I would have wondered if its different in colder soils. line 664-667 - What about other proxies like pH?

*Unfortunately, the role of pH and other soil factors could not be assessed with our datasets*

line 668-669 - Do you have other soil parameters that could be used to fine tune the multiple regressions?

*Not for the full datasets*
I find this manuscript a well written and valuable study addressing the vulnerability of permafrost carbon to climate change and the permafrost carbon feedback. It seems, however, that the authors assume recent permafrost warming as general knowledge. At the beginning of the introduction they could better provide proofs and estimates of this ground warming process and the relationship to the atmosphere. Recent publications on permafrost warming should enable also a direct comparison to some of the considered study sites.

*AR15: we will add a paragraph to the introduction stating the conclusions of the most recent IPCC report (2018), which identifies the permafrost carbon feedback as a key uncertainty (together with peatlands, which also are widespread in the northern permafrost region) in assessments to keep global warming under 1.5/2 °C. Furthermore, the urgency of additional research is highlighted by the fact that most permafrost in the northern circumpolar region has already experienced warming in recent decades (Biskaborn et al., 2019)*

*Additional reference:*

*Biskaborn, B. et al., 2019. Permafrost is warming at a global scale. Nature Communications, 10: 264. doi: 10.1038/s41467-018-08240-4*

---

## Author Response (AR1)

Stockholm, 31-10-2019

Re.: Revised version of manuscript BG-2019-89

Dear Associate Editor,

I have uploaded the revised version of the manuscript as a word file of the main text, with all changes highlighted in track changes (including tables and figures), as well as a pdf of the revised supplement.

I believe we have addressed all concerns made in the reviews and the one interactive comment that we suggested in our Author's response.

There were some technical/formatting changes requested by the reviewers, most significantly:

1) To remove Figure 2, because these regressions were also included in Figure 3, etc
2) To prepare a Table S2 for Supplementary Materials that summarizes landscape representation in each experiment/study area
3) To move paragraphs in results to methods, particularly the ones explaining the division of the datasets into landscape classes (and their subdivisions in the PAGE21 experiment)
4) To add number of samples (and, in some cases, legends) in the Figures

As for the text, following the main concerns raised in the reviews, the most important changes are:

5) Expanded method description for the CryoCarb experiment
6) Expanded discussion

Detailed summary of most important changes:

| | |
|---|---|
| Line 3: | Added co-author |
| L54-58 | Added statements |
| L182-183 | Added statement |
| L195-234 | Additional details for CryoCarb experiment |
| L285-287 | Added Table S2 |
| L288-299 | From results to methods |
| L317-327 | From results to methods |
| L352 + L362 | Added number of samples for each regression |
| L385 | Removed Figure 2 |
| L540 | Small error in slope value for PAGE21 Pt, corrected 0.20 to 0.24, 0.14 to 0.17 |
| L691-698 + L708-749 | Added discussion paragraphs |
| L840+ | Added references |
| | |
| Figures 2, S3-S5 | Added number of samples |
| Figures 3 and 4 | Added legends and number of samples, Change in symbol and color regression line in Fig. 3d |

We hope you find these revisions adequate.

Kind regards, Peter Kuhry

[revised manuscript text omitted]

Figure 2

a b

Figure 3

**a. Eolian (n=76)**

[Figure]

**c. Mineral (n=51)**

[Figure]

**b. Alluvial (n=82)**

[Figure]

**d. Wetland (n=24)**

[Figure]

**Figure 4**

[Figure]

a

[Figure]

b

[Figure]

**Figure 5**

[Figure]

[Figure]